

# New age constraints reveal moraine stabilization thousands of years after deposition during the last deglaciation of western New York, USA

Karlee K. Prince[1], Jason P. Briner[1], Caleb K. Walcott[1], Brooke M. Chase[1], Andrew L. Kozlowski[2], Tammy M. Rittenour[3], Erica P. Yang[1,4]

[1] Department of Geology, University at Buffalo, 126 Cooke Hall, Buffalo, NY 14260, USA
[2] New York State Geological Survey, New York State Museum, 222 Madison Ave, Albany, NY 12230, USA
[3] Department of Geoscience, Utah State University, 4505 Old Main Hill, Logan, UT 84322, USA
[4] Oak Ridge Institute of Science and Education, 1299 Bethel Valley Road, Oak Ridge, TN, 37830 USA

*Correspondence to:* Karlee K. Prince (karleepr@buffalo.edu)



**Abstract.** The timing of the last deglaciation of the Laurentide Ice Sheet in western New York is poorly constrained. The lack of direct chronology in the region has led to a provocative hypothesis that the Laurentide Ice Sheet re-advanced to near its Last Glacial Maximum terminal position in western New York at ~13 ka, which challenges long-standing datasets. To address this hypothesis, we obtained new chronology from the Kent (terminal) and Lake Escarpment (first major recessional) moraines using radiocarbon ages in basal sediments from moraine kettles supplemented with two optically stimulated luminescence ages. The two optically stimulated luminescence ages date the Kent (terminal) position to $19.8 \pm 2.6$ and $20.6 \pm 2.9$ ka. Within the sediment cores from both moraines, the lowest reliable radiocarbon ages range from 15,000-15,400 to 13,600-14,000 cal yr BP. Below these dated levels is sedimentologic evidence of an unstable landscape during basin formation; radiocarbon ages from these lowest sediments are not in stratigraphic order and date from 19,350-19,600 to 14,050-14,850 cal yr BP. The oldest radiocarbon age of 19,350-19,600 cal yr BP – from a rip-up clast – suggests ice-free conditions at that time. We interpret that the 5 kyr lag between the optically stimulated luminescence ages and the lowest reliable radiocarbon ages is the result of persistent buried ice in ice-cored moraines until ~15 to 14 ka. The cold conditions associated with Heinrich Stadial 1 may have enabled the survival of ice-cored moraines in permafrost until after 15 ka, and in turn, climate amelioration during the Bølling Period (14.7 – 14.1 ka) may have initiated landscape stabilization. This model potentially reconciles the sedimentological and chronological evidence underpinning the provocative re-advance hypothesis, which instead could be the result of moraine instability during the Bølling-Allerød periods (14.7 – 13 ka). Age control for future work should focus on features that are not dependent on local climate.

## 1 Introduction

Much glacial research over the last century has focused on the style and timing of Laurentide Ice Sheet (LIS) recession from the Great Lakes region of North America following the Last Glacial Maximum (LGM, 26-19 ka; Dalton et al., 2020; Dyke, 2004; Fairchild, 1909). Well constrained ice sheet chronologies are necessary to constrain the timing of meltwater re-routing events from ice-dammed lakes that occupied the Great Lakes basins during the last deglaciation (Barth et al., 2019; Calkin and Feenstra, 1985; Leydet et al., 2018; Porreca et al., 2018; Rayburn et al., 2007), as these events are hypothesized to have had significant climatic impacts (Broecker et al., 1989; Donnelly et al., 2005). Models that attempt to understand past climate change (Osman et al., 2021), ice sheet sensitivity (Briner et al., 2020), and atmospheric organization (Löfverström et al., 2014; Tulenko et al., 2020) all require paleo ice sheet configurations. Therefore, well-defined ice sheet retreat chronologies are critical for understanding dynamics and forcings within the late glacial climate system.

Despite the critical need for precise chronologies of ice margin retreat of the LIS in the Great Lakes region, ice margin reconstructions in western New York lack detailed age control. Here, there are no local ages on the terminal moraine and few from the recessional moraines (Muller and Calkin, 1993), leaving the deglacial chronology to be largely based on correlations with dated moraines and proglacial shorelines to the west in Ohio and to the east in New York (Fullerton, 1980; Ridge, 2003). These correlations suggest that the western New York Kent (terminal) and Lake Escarpment (recessional) moraines date to ~20 and 17 ka, respectively (Fig. 1). However,



Young et al. (2020) recently interpreted new radiocarbon ages from western New York to support a significant
re-advance of the LIS at ~13 ka that overtopped the Lake Escarpment Moraine and nearly reached the Kent Moraine
(Fig. 1). In contrast to Young et al.'s (2020) reconstruction, most literature places the LIS margin north of Lake
Ontario at this time (Dalton et al., 2020; Muller and Calkin, 1993; Terasmae, 1980; and references therein), with the
drainage of Glacial Lake Iroquois occurring at ~13 ka (Fig. 1; Cronin et al., 2012; Lewis and Anderson, 2019;
Rayburn et al., 2005). If a re-advance of the scale hypothesized by Young et al. (2020) occurred (henceforth referred
to as the 'Allerød re-advance hypothesis'), we would need to revisit many regional deglaciation chronologies.

57    To further constrain moraine ages in western New York and to test the Allerød re-advance hypothesis, we

obtained 23 new macrofossil-based radiocarbon ages from five sediment cores collected on the Kent Moraine, and
18 new macrofossil-based radiocarbon ages from two sediment cores on the Lake Escarpment Moraine. The Lake
Escarpment Moraine is within the extent of the proposed re-advance, so if basal ages from sites on this moraine
pre-date ~13 ka, and the subsequent stratigraphy shows no evidence of a re-advance, then the evidence would refute
the Allerød re-advance hypothesis. Conversely, basal radiocarbon ages that post-date ~13 ka, and/or evidence that
the sediment stratigraphy is interrupted at ~13 ka, would support an Allerød re-advance. Additionally, we obtained
two optically stimulated luminescence (OSL) ages from kame delta sediments associated with deposition of the Kent
Moraine to provide a more complete understanding of deglaciation. Our results provide new chronological
constraints in the western New York data gap, and do not support the ~13 ka re-advance proposed by Young et al.
(2020). Rather, our data support a model of initial moraine deposition followed by thousands of years before kettle
basin formation and final moraine stabilization.



**Figure 1. Map depictions of the deglaciation of the eastern Great Lakes after the Last Glacial Maximum. Black line is the Kent Moraine, modified from Dalton et al. (2020), the 'Pennsylvania Department of Conservation and Natural Resources Late Wisconsin Glacial Border' (https://www.pasda.psu.edu), and the 'Quaternary Geology 500K - Glacial Boundary of Ohio' (https://gis.ohiodnr.gov). Dark gray line is the 17 ka ice margin from Dalton et al. (2020) which depicts the Lake**



Escarpment Moraine. Light gray line is the 15 ka ice margin from Dalton et al. (2020) which depicts the Marilla Moraine.
Glacial Lake Maumee and Whittlesey are included for general reference, and derived with shoreline elevations (Fisher et
al., 2015). White line is the 13 ka ice margin from Dalton et al. (2020) and we estimated Glacial Lake Iroquois using Bird
and Kozlowski (2016). Red dashed line depicts a hypothesized ice sheet configuration to explain the hypothesis presented
in Young et al. (2020). Note that the LIS would dam a pro-glacial lake in the Lake Erie basin and overrun several moraine
belts, including the Lake Escarpment Moraine. DEM from U.S. Geological Survey's Center for Earth Resources
Observations and Science (EROS).

## 82  2 Geologic Setting

The Kent Moraine in western New York is correlated to the Kent Moraine in northwest Ohio, the Olean

Moraine in Pennsylvania, the Harbor Hill Moraine in New Jersey, and the Martha's Vineyard Moraine in
Massachusetts (Fig. 1; Balco et al., 2002; Fullerton, 1980; Muller and Calkin, 1993; Stanford et al., 2020). Retreat
from the LGM moraine in these adjacent regions is dated to $19.8 \pm 0.4$ ka in Ohio (Glover et al., 2011), $25.2 \pm 2.1$ ka
(Corbett et al., 2017) and 23,200-23,750 cal yr BP in New Jersey (Stanford et al., 2020), and $25.5 \pm 0.4$ ka in
Massachusetts (Balco et al., 2009; Balco et al., 2002). Therefore, we infer that the Kent Moraine in western New
York was likely deposited sometime between 25 and 20 ka.

The first major moraine belts deposited after the maximum LGM position were the Ashtabula Moraine in

Ohio and northwest Pennsylvania, the Lake Escarpment Moraine in western New York, and Valley Heads moraines
in central New York (Fig. 1; Fullerton, 1980; Muller and Calkin, 1993). During this ice position, Glacial Lake
Maumee occupied the Lake Erie basin, and is dated to 17 - 16 cal ka BP by radiocarbon dating at the paleo-outlet
and OSL dating of strandlines (Calkin and Feenstra, 1985; Eschman and Karrow, 1985; Fisher et al., 2015). Ridge
(2003) tied the outer and inner Valley Heads moraines to the New England Varve Chronology, placing these
moraines at 17,200 and 16,200 cal yr BP, respectively. Kozlowski et al. (2018) report basal ages of 14,300-14,900
and 14,200-14,850 cal yr BP from basins within the outer Valley Heads limit. These ages are younger than previous
estimates, leading Kozlowski et al. (2018) to suggest the moraine may have been re-occupied. Calkin and
McAndrews (1980) report minimum-limiting radiocarbon ages of 13,750-15,250 cal yr BP from wood
stratigraphically above outwash sands from Nichols Brook in western New York (Fig. 2). Muller and Calkin (1993)
extrapolated their ages to estimate ~17,600 cal yr BP for the emplacement of the outwash.
Following the deposition of the Lake Escarpment Moraine, Glacial Lakes Whittlesey and Warren occupied
the Lake Erie basin between 16 and 14 ka (Fig. 1; Fullerton, 1980; Muller and Calkin, 1993). The lowering of
Glacial Lake Whittlesey to Glacial Lake Warren is dated to 14,150-15,550 cal yr BP at Winter Gulf in western New
York (Fig. 2; Calkin and McAndrews, 1980), and Warren strandlines in northwest Ohio have been dated to $14.2 \pm$
1.3 ka (Higley et al., 2014) and $14.1 \pm 1.0$ ka in (Campbell et al., 2011). These proglacial lake chronologies provide
unambiguous minimum age constraints of >15 ka for the deposition of the Lake Escarpment Moraine.
The LIS continued its northward retreat and formed Glacial Lake Iroquois from 14.7 to 13.0 ka in the Lake
Ontario basin (Fig. 1; Muller and Calkin, 1993; Muller and Prest, 1985; Teller, 2003). The switch of the Glacial
Lake Iroquois spillway from the Mohawk River valley to the lower outlet at Covey Hill is constrained between
13,200 and 13,000 cal yr BP by numerous radiocarbon constraints from the pre- and post-flood histories of Lake



Vermont and Lake Iroquois (Lewis and Anderson, 2019; Rayburn et al., 2007; Richard and Occhietti, 2005).
Similarly, the formation of the Champlain Sea occurred between 13,100 and 12,700 cal yr BP, which post-dates the
final draining of Glacial Lake Iroquois and requires an ice margin north of the Lake Ontario outlet (Cronin et al.,
2012; Rayburn et al., 2011). Collectively, this ice recession chronology is at odds with the Allerød re-advance
hypothesis, with its significant LIS advance across the Lake Ontario basin and to near the terminal moraine in
western New York ~13 ka (Fig. 1; Young et al., 2020).

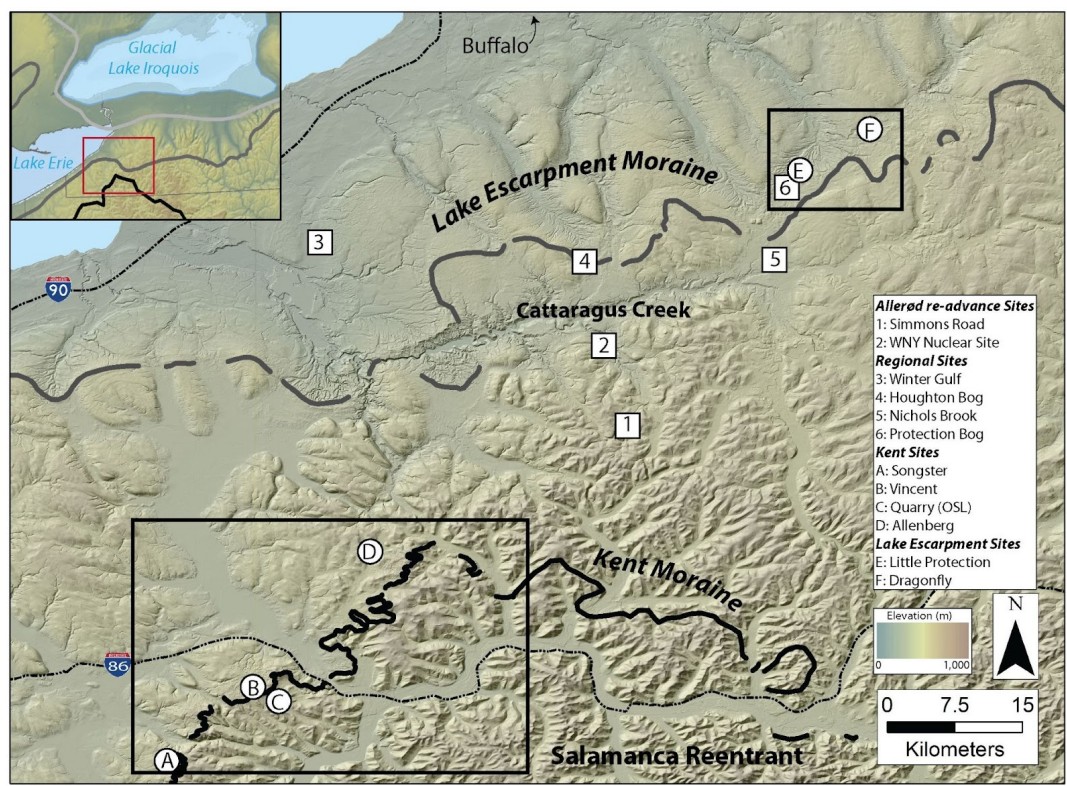


**Figure 2. Study sites in relation to previously published work. Black and gray lines are the same as in Fig. 1. Squares 1 and 2 depict hypothesized sites overrun by the Allerød re-advance at 13 ka (Young et al., 2020). Circles A-D are our sites on the Kent Moraine. Circles E and F are our sites on the Lake Escarpment Moraine. Squares 3-6 are Winter Gulf and Nichols Brook (Calkin and McAndrews, 1980), and Houghton and Protection Bog (Miller, 1973). The two black boxes show the extent of the maps in Fig. 3. DEM from U.S. Geological Survey's Center for Earth Resources Observations and Science (EROS).**


**3 Methods**
**3. 1 Sediment cores**

Our primary approach for constraining the timing of deglaciation and testing the Allerød re-advance

hypothesis was obtaining basal sediment ages from kettles within the Kent and Lake Escarpment Moraines. Newly



available light detection and ranging (LiDAR)-based bare-Earth 1-m digital elevation models (DEMs) enabled us to
identify natural kettle basins (Fig. 3). Typically, moraines in western New York have both single ridges where the ice
sheet abutted higher topography, and hummocky morainal belts that contain numerous kettle basins. Kame deltas
exist in places where the ice sheet dammed adjacent river valleys. The hummocky nature of most moraines indicates
that the moraines were ice-rich when deposited (Fig. 3).

We collected sediment cores from kettles that presently range from bogs to wetlands. We cored five sites on
the Kent Moraine referred to as the Vincent-1 (core name: 20VIN1), Vincent-3 (20VIN3), Vincent-4 (20VIN4),
Songster (21SONG1), and Allenberg (15ABB7) sites (Table 1, Fig. 3), and two sites on the Lake Escarpment
Moraine referred to as the Little Protection (21LPB1) and Dragonfly (13DFK1) sites (Table 1, Fig. 3). All sites are
within hummocky moraine.

We determined basin depocenters using thin steel rods to measure the depth of the organic sediment infill.
We used Livingstone- and Russian Peat-style corers to collect organic-rich sediment infill, and a manual percussion
GeoProbe system to collect the underlying stiff, minerogenic sediments. From some sites, our sediment cores
extended from the present surface to mineral-rich sediments below the organic-sediment infill; from others, our
sediment cores began and ended at depth, spanning the organic-to-mineral sediment contact and downward until we
penetrated coarse deposits (Table 1). We returned and cored the Vincent-1 and -4 sites multiple times to collect the
entire sequence.

We split, imaged, and generated downcore data on all sediment cores at the University at Buffalo. We
measured magnetic susceptibility in contiguous 1 cm intervals using a Bartington MS2E High Resolution Surface
Scanning Sensor scanner connected to a Bartington MS2 Magnetic Susceptibility Meter to assess the minerogenic
content. We calculated loss-on-ignition (LOI) percent by burning ~1 cm³ of sediment in a Thermolyne Muffle
Furnace at successively higher temperatures for water (105°C), organic carbon (550°C), and carbonate (950°C)
content to help characterize the sediment units and depositional setting (Heiri et al., 2001; Last and Smol, 2001). To
calculate composite core length, we spliced together overlapping sediment sections using visual lithologic changes
and magnetic susceptibility measurements. We volumetrically sampled portions of the Little Protection sediment
cores to determine sediment bulk density.

We use radiocarbon dating of macrofossils for age control (Table 2). The sediments are organic-rich in the
upper portions of the cores and are organic-poor in the lower sections. Where available, we picked full plant
macrofossils. We picked macrofossils that were from the center of the sediment core and demonstrably in-situ. In
macrofossil-devoid sections, we wet sieved sediment with deionized water to isolate and combine the largest
macrofossil fragments for dating. We attempted to identify macrofossils, but some macrofossil fragments were small
and unidentifiable (Table 2). We rinsed samples with deionized water, freeze-dried them, and sent samples to the
National Ocean Sciences Accelerator Mass Spectrometry (NOSAMS) or the Keck Lab at the University of
California Irvine (KCCAMS) for radiocarbon analysis. We submitted untreated macrofossils, therefore the facilities
conducted acid-base-acid (ABA) pretreatments, converted samples to graphite, and ran them on the AMS (Elder et
al., 2019; Olsson, 1986; Pearson et al., 1997; Shah Walter et al., 2015; Vogel et al., 1984).



We report the entire 2σ age range and round ages according to Stuiver and Polach (1977) (Table 2)  We
calibrated all the radiocarbon results using Calib8.1 with the IntCal20 dataset (Reimer et al., 2020; Stuiver and
Reimer, 1993). $\delta^{13}C$ measurements were measured on a split of the $CO_2$ gas generated from each sample on an
isotope-ratio mass spectrometer. Uncertainties in the $\delta^{13}C$ from both labs are <0.1‰. We report $\delta^{13}C$ values as ‰
VPDB.

Table 1: Site location, core lengths, and ownership.

| Site Name | Core Name | Latitude (DD) | Longitude (DD) | Elevation (m asl) | Site Length (m) | Core Top (m bg) | Core Bottom (m bg) | Property Ownership |
|---|---|---|---|---|---|---|---|---|
| Vincent 1 | 20VIN1 | 42.109 | -79.000 | 596 | 145.0 | 0.0 | 6.6 | Vincent Family |
| Vincent 3 | 20VIN3 | 42.110 | -78.999 | 593 | 39.0 | 1.5 | 2.9 | Vincent Family |
| Vincent 4 | 20VIN4 | 42.109 | -78.999 | 594 | 81.0 | 3.1 | 5.4 | Vincent Family |
| Songster | 21SONG1 | 42.040 | -79.079 | 581 | 172.0 | 4.1 | 4.8 | Songster Family |
| Allenberg | 15ABB7 | 42.252 | -78.883 | 524 | 321.0 | 8.0 | 14.6 | Buffalo Audubon Society |
| Little Protection | 21LPB1 | 42.621 | -78.463 | 440 | 228.0 | 0.0 | 8.1 | Erie County Parks Dept. |
| Dragonfly | 13DFK1 | 42.679 | -78.386 | 450 | 117.0 | 0.0 | 7.3 | Buffalo Audubon Society |
| Corbett Hill | - | 42.114 | -78.946 | 530 | - | - | - | JMI Corbett Hill Gravel |

DD: Decimal Degrees
asl: Above sea level
bg: Below ground






Figure 3. Site maps of the sediment core locations. 1-m bare-Earth DEM hillshade from https://data.gis.ny.gov/ with the
Kent (black) and Lake Escarpment (gray) moraines. Open yellow circles depict study site location and yellow lines



associate each site location with a site map. Figure 4 contains the site map for the open yellow circle with no associated site
map. The filled circles indicate the type of coring device used in each site and the coring location. The filled yellow circles
depict where we used a Livingstone. The filled red circles depict where we used a Russian Peat Corer. The filled
semi-circles indicate where we used a Livingston or Russian Peat Corer in the soft sediment infill and then used the
GeoProbe in the stiff minerogenic sediment.

**3.2 Optically stimulated luminescence dating**

We collected sediment samples for OSL dating from topset beds within an ice-contact delta deposit

associated with the Kent Moraine to determine when the LIS was present at this location (Fig. 3 & 4). Our sample
location was Corbett Hill Gravel Quarry, an active aggregate quarry that exposes large sedimentary sequences
indicative of a proglacial delta. The sediments consisted of cobble-rich foreset beds overlain by ~3 m of
near-horizontal topset beds. We collected sand samples for OSL dating from the topset sequence ~2.1 m below the
delta surface. We created a fresh exposure of the topset beds with an excavator, exposing alternating layers of
gravels and coarse sands, with lenses of medium/fine-sand and silt. We collected two samples for OSL dating in
fine-sand lenses in 5.1 x 25.4 cm (2 x 10 inch) aluminum tubes after clearing back outer sediments (Fig. 4). Samples
for water content and dose rate determination were collected from surrounding sediments.

We processed the samples at the Utah State University Luminescence Laboratory for small aliquot OSL

dating of fine-grained quartz sand (Table 3, ATable 1). First, we purified samples to 150-250 μm quartz sand using
wet sieving, and chemical treatment with 10% hydrochloric acid to remove carbonates, 5% peroxide to remove
organics, 2.72 g/cm$^3$ sodium polytungstate to remove heavy minerals and 48% hydrofluoric acid to remove feldspars
and etch the quartz grains. We analyzed small aliquots of quartz (0.4 to 1 mm diameter of sand mounted on disk,
~10-20 grains) on Risø DA-20 readers, using the single-aliquot regenerative-dose (SAR) protocol (Murray and
Wintle, 2000). We analyzed 42 aliquots for sample 21SICK-01 and 37 for sample 21SICK-02, of which we used 21
and 23 aliquots for age calculations, respectively (Fig. A1 & A2). Aliquots were rejected from age calculation if
they showed signal depletion with infrared stimulation indicating feldspar contamination (0-12 aliquots), poor
recycling of a repeat point (greater than 80% difference between repeat points, 7-8 aliquots), high recuperation of a
zero-dose point ( >10% of the Natural signal, 0-6 aliquots), extrapolation of the equivalent dose beyond the
dose-response curve (0-2 aliquots) and poor dose-response curve fit (0-3 of aliquots). We applied a minimum age
model (MAM) to the samples to calculate our equivalent does ($D_E$; Grays; Gy, Fig. A1 & A2), as used by similar
studies on LIS glaciofluvial terraces elsewhere in the northern United States (Rittenour et al., 2015).

We determined the dose rate for OSL age calculation based on U, Th, K, and Rb concentrations from the

surrounding sediments using inductively coupled plasma-mass spectrometry and atomic emission spectrometry.
Using the conversion factors of Guérin et al. (2011), we converted elemental concentrations to dose rate. The
contribution of cosmic radiation was based on sample depth, elevation and latitude following Prescott and Hutton
(1994). We also determined water content by measuring the mass of the samples before and after desiccation. With
these three factors, we were able to calculate environmental dose rates (Gy/kyr). Our reported OSL ages are simply
the $D_E$ (determined with the MAM) divided by the dose rate with 1σ standard error (Table 3). We report ages with
1σ uncertainty (Table 3).



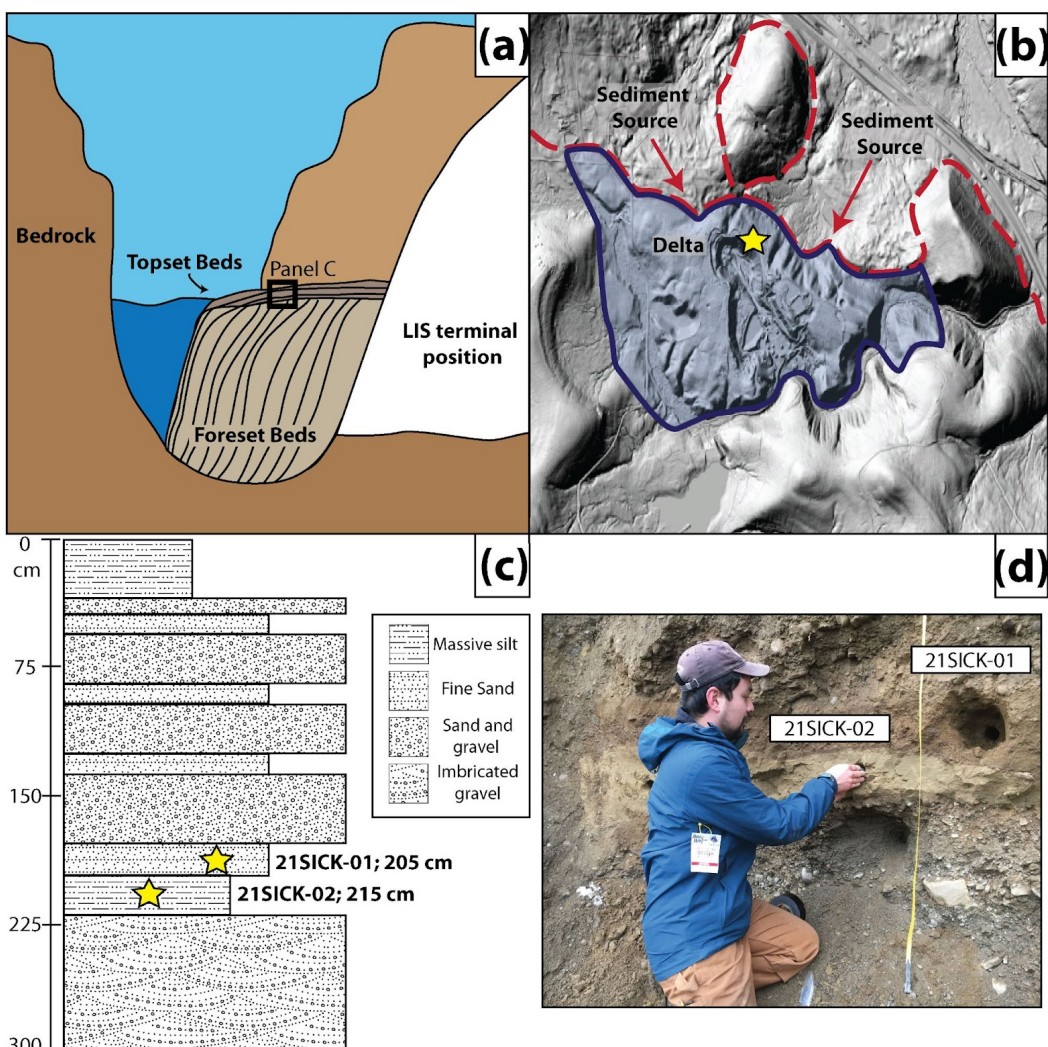

**Figure 4. Panel A) is a schematic of the kame delta creation. The LIS dammed a lake and deposited the delta outboard of the Kent Moraine. B) is a 1-m DEM hillshade showing the kame delta outboard of the Kent Moraine (within the open yellow circle in Fig. 3). Red dashed line depicts the extent of the Kent Moraine. Red arrows depict the sediment source for the delta. Blue line and shading depicts the delta deposit. Yellow star on the side of the active quarry shows our sampling site. C) shows a stratigraphic column of the topset beds. We use the FGDC Digital Cartographic Standard for Geologic Map Symbolization (U.S. Geological Survey). Yellow stars show our sampling location. D) is a field photo of the stratigraphic section showing the location of our two samples.**



## 4 Results

### 4.1 Stratigraphy

The sediment cores contain three stratigraphic units: a basal unit (Unit 1) of diamicton, an intermediate unit (Unit 2) dominated by silt and sand, and an upper unit (Unit 3) of organic-rich silt and peat (Fig. 5). We only recovered Unit 1 in 20VIN1 using the Geoprobe system (Fig. 5). Unit 1 is a gray massive pebbly diamicton with a silty matrix. The contact with Unit 2 is sharp.

We collected varying thicknesses of Unit 2 (Fig. 5). Unit 2 is mineral-rich layers with complex stratigraphy and sparse macrofossil fragments. In 20VIN1, Unit 2 is layered gray sand and silt that grades to alternating massive brown and gray silt (Fig. 5). In 20VIN3, Unit 2 begins as gray silt, transitions to a light brown silt, and is topped by gray clay. In 20VIN4, Unit 2 contains alternating layers of pebbly diamicton (with some clasts up to 5 cm long) and silty clay. The contacts between the layers are sharp and one is undulating. In 21LPB1, Unit 2 begins with 2 cm of gray silty gravel before a sharp contact with massive, oxidized sand and gravel. Above this is a sharp transition to alternating layers of gray silt, silty gravel, and sand; these layers have sharp and sometimes undulating contacts. That is overlain by massive gray clay. In 13DFK1, Unit 2 is gray silt. The contact between Unit 2 and 3 is sharp in all cores.

Unit 3 is organic-rich silt (lower Unit 3) and peat (upper Unit 3) that spans from the mineral layers of Unit 2 to the top of each sediment sequence. The transition from organic-rich silt to peat is sharp in all cores. In 20VIN1, within the initial sediments of Unit 3, there are three layers of gray silt and an inclusion of gray clay that are identical to the sediment of Unit 2. Similarly, 20VIN3 has a layer of gray silt within the initial organic-rich silt. The organic-rich silt and peat have high organic carbon content and large macrofossils are common.

To address the Allerød re-advance hypothesis and seek evidence of whether the kettle sediments were overridden, we measured dry bulk density at 1-cm-resolution through the time interval of hypothesized re-advance in our Little Protection site core (21LPB1). The bulk density decreases from 1.55 g/cm$^3$ to 0.42 g/cm$^3$ in the transition from Unit 2 to 3. (Fig. 5) The density decreases due to the transition from minerogenic silt to organic-rich silt and remains below 0.42 g/cm$^3$ into Unit 3.







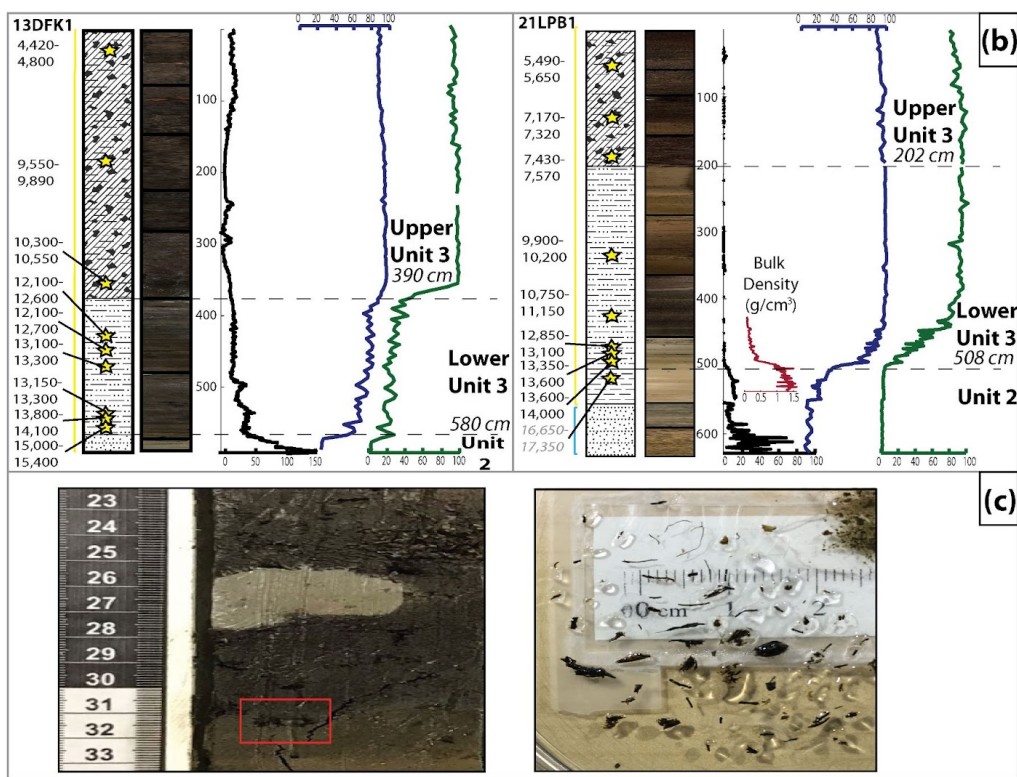

**Figure 5. Panel A) has the sediment core stratigraphy from the Kent Moraine sites, and B) has the sediment core stratigraphy from the Lake Escarpment Moraine sites. We show sediment texture next to the core images using the FGDC Digital Cartographic Standard for Geologic Map Symbolization (U.S. Geological Survey). We plot magnetic susceptibility (CGS; black line), water content (weight %; blue line), and organic content (weight %, green line) by composite depth (cm). The colored line next to the stratigraphic column depicts if we used the Russian Peat Corer (red), Livingstone Corer (yellow), or GeoProbe (blue). Yellow stars indicate radiocarbon ages (2σ uncertainty cal yr BP). We used gray text and italics for radiocarbon ages we suspect have hardwater contamination. C) is a close-up image of the inferred gray clay and macrofossil-rich rip-up clasts in the transition from Unit 2 to 3 in 20VIN1 (shown in red box). The black box has post-sieve macrofossils from the rip-up clast in the red box.**

## 4.2 Sediment core chronology

The stratigraphically lowest ages, or basal ages, from the Kent Moraine range from 15,050-15,550 to 13,800-14,050 cal yr BP (Table 2; Fig. 5). For 20VIN3, 20VIN4, and 21SONG1, the basal ages cluster around 14,700 cal yr BP. The basal ages from the Lake Escarpment Moraine are 15,000-15,400 and 16,650-17,350 cal yr BP. The basal ages are not the oldest ages, however. 20VIN1 has a basal age of 15,050-15,550 cal yr BP, yet is stratigraphically below other ages from Unit 2 of 15,650-15,900, 15,800-16,150, and 16,050-16,300 cal yr BP.



Furthermore, 110 cm higher than the basal age, there is an inclusion of macrofossils within Unit 2 that was dated
twice and yields two radiocarbon ages of 19,350-19,600 and 14,050-14,850 cal yr BP; combined macrofossils from
the surrounding sediment produce an age of 14,300-15,050 cal yr BP. In 20VIN3, the basal age is 14,350-15,150 cal
yr BP, yet combined macrofossils higher in the core, at the Unit 2/3 boundary, produce an age of 15,350-15,650 cal
yr BP.





Table 2: Radiocarbon dates for each study site. Listed by depth.

| Lab Code | Depth (cm) | Fraction Modern | Fraction Modern Error | Mass (mg) | Material Dated | $\delta^{13}C$‰ | $^{14}C$ (yr) | $^{14}C$ error (yr) | 2 σ age range (cal BP) | | Median (cal BP) |
|---|---|---|---|---|---|---|---|---|---|---|---|
| **Vincent-1 (20VIN1)** | | | | | | | | | | | |
| OS-164770 | 27.3 | 0.3854 | 0.0015 | 5.8 | Wood | -26.50 | 7,660 | 30 | 8,390 | 8,540 | 8,440 |
| OS-164771 | 67.3 | 0.2441 | 0.0015 | 5.8 | Twig | -28.67 | 11,350 | 50 | 13,100 | 13,300 | 13,250 |
| OS-164772 | 109.5-111.0 | 0.2263 | 0.0016 | 75.7 | *Picea* seeds | -22.96 | 11,950 | 55 | 13,600 | 14,050 | 13,850 |
| OS-164773 | 145.0 | 0.2105 | 0.0016 | 2.3 | Unidentifiable | -24.32 | 12,500 | 60 | 14,300 | 15,050 | 14,700 |
| UCIAMS-239749 | 145.2 | 0.1342 | 0.0007 | 2.4 | Moss, unidentifiable | -28.30 | 16,135 | 45 | 19,350 | 19,600 | 19,500 |
| OS-164808 | 145.2 | 0.2169 | 0.0019 | 2.4 | Moss, unidentifiable | -26.79 | 12,300 | 70 | 14,050 | 14,850 | 14,300 |
| UCIAMS-239748 | 181.0-182.5 | 0.1949 | 0.0006 | 6.8 | *Drepanocladus, Paludella squarrosa, Potomegeton,* unidentifiable | -19.40 | 13,135 | 30 | 15,650 | 15,900 | [a]15,750 |
| UCIAMS-239746 | 185.5-188.5 | 0.1875 | 0.0006 | 2.1 | Moss, *Potomegeton,* unidentifiable | -14.60 | 13,450 | 25 | 16,050 | 16,300 | [a]16,200 |
| UCIAMS-239745 | 239.0-241.5 | 0.1910 | 0.0009 | 2.5 | Unidentifiable | NA | 13,300 | 40 | 15,800 | 16,150 | [a]16,000 |
| OS-162874 | 255.0 | 0.2029 | 0.0020 | 2.5 | Moss, unidentifiable | -26.60 | 12,800 | 80 | 15,050 | 15,550 | 15,300 |
| **Vincent-3 (20VIN3)** | | | | | | | | | | | |
| UCIAMS-239753 | 34.5-35.5 | 0.1988 | 0.0006 | 25.8 | *Chara,* unidentifiable | -15.30 | 12,980 | 25 | 15,350 | 15,650 | [a]15,550 |
| OS-162873 | 146.5-152.0 | 0.2100 | 0.0020 | 2.1 | Unidentifiable | -26.56 | 12,550 | 75 | 14,350 | 15,150 | 14,850 |
| **Vincent-4 (20VIN4)** | | | | | | | | | | | |
| UCIAMS-239752 | 17.0-18.0 | 0.2126 | 0.0007 | 5.1 | Beetle wing, *Cladocera, Chara,* unidentifiable | -24.90 | 12,435 | 30 | 14,300 | 14,900 | 14,550 |
| UCIAMS-239751 | 87.0-88.0 | 0.2127 | 0.0010 | 39.2 | Unidentifiable | NA | 12,435 | 40 | 14,300 | 14,900 | 14,550 |
| UCIAMS-239750 | 97.5-98.8 | 0.2150 | 0.0011 | 6.3 | Unidentifiable | NA | 12,350 | 45 | 14,150 | 14,850 | 14,400 |
| OS-162875 | 174.0-175.0 | 0.2120 | 0.0019 | 2.0 | Twig | -27.97 | 12,450 | 75 | 14,250 | 15,000 | 14,600 |
| **Songster (21SONG1)** | | | | | | | | | | | |
| OS-160884 | 39.3 | 0.2107 | 0.0015 | 6.4 | Bark (likely *Picea*) | NA | 12,500 | 55 | 14,350 | 15,050 | 14,700 |
| **Allenberg (15ABB7)** | | | | | | | | | | | |
| OS-123347 | 971.0 | 0.8877 | 0.0019 | NA | Not identified | -25.96 | 955 | 20 | 795 | 920 | 850 |

*continued*







| Sample ID | Mass (mg) | Fraction Modern | Error | %C | Material | δ13C | 14C age | ± | cal BP (min) | cal BP (max) | cal BP (median) |
|---|---|---|---|---|---|---|---|---|---|---|---|
| OS-123426 | 1178.0 | 0.4580 | 0.0018 | NA | Not identified | -27.09 | 6,270 | 30 | 7,030 | 7,270 | 7,210 |
| OS-123427 | 1295.0 | 0.2607 | 0.0020 | NA | Not identified | -26.82 | 10,800 | 60 | 12,700 | 12,850 | 12,750 |
| OS-123348 | 1456.0 | 0.2227 | 0.0012 | NA | Not identified | -24.62 | 12,050 | 40 | 13,800 | 14,050 | 13,900 |
| **Little Protection (21LPB1)** | | | | | | | | | | | |
| OS-163424 | 53.7 | 0.5464 | 0.0014 | 4.6 | Wood | -24.58 | 4,860 | 20 | 5,490 | 5,650 | 5,590 |
| OS-163425 | 141.0 | 0.4549 | 0.0017 | 15.6 | Wood | -28.34 | 6,330 | 30 | 7,170 | 7,320 | 7,250 |
| OS-163426 | 198.5 | 0.4376 | 0.0013 | 39.7 | Wood | -27.97 | 6,640 | 25 | 7,430 | 7,570 | 7,530 |
| OS-163427 | 320.5 | 0.3295 | 0.0012 | 5.4 | *Potomegeton* | -17.55 | 8,920 | 30 | 9,900 | 10,200 | [a]10,050 |
| OS-163428 | 423.0 | 0.3034 | 0.0015 | 5.8 | Seed pod | -27.98 | 9,580 | 40 | 10,750 | 11,150 | 10,950 |
| OS-163517 | 472.2 | 0.2512 | 0.0016 | 5.5 | *Picea* cone | -25.65 | 11,100 | 50 | 12,850 | 13,100 | 13,000 |
| OS-163500 | 481.0 | 0.2346 | 0.0017 | 32.9 | Wood | -26.95 | 11,650 | 60 | 13,350 | 13,600 | 13,500 |
| OS-163501 | 493.0 | 0.2277 | 0.0015 | 3.0 | Wood | -27.27 | 11,900 | 55 | 13,600 | 14,000 | 13,750 |
| OS-163429 | 511.0 | 0.1749 | 0.0023 | 11.0 | Fish bone | -26.48 | 14,000 | 110 | 16,650 | 17,350 | [a]17,000 |
| **Dragonfly (13DFK1)** | | | | | | | | | | | |
| OS-106743 | 25.2 | 0.6025 | 0.0031 | NA | Twig | -22.80 | 4,070 | 40 | 4,420 | 4,800 | 4,560 |
| OS-106745 | 194.9 | 0.3381 | 0.0016 | NA | Moss stems | -26.60 | 8,710 | 40 | 9,550 | 9,890 | 9,650 |
| OS-106746 | 362.4 | 0.3157 | 0.0017 | NA | Moss stems | -25.00 | 9,260 | 45 | 10,300 | 10,550 | 10,450 |
| OS-133658 | 431.5 | 0.2729 | 0.0015 | NA | Leaf | -25.70 | 10,450 | 45 | 12,100 | 12,600 | 12,350 |
| OS-106747 | 453.5 | 0.2704 | 0.0017 | NA | Twig | -25.90 | 10,500 | 50 | 12,100 | 12,700 | 12,550 |
| OS-133659 | 482.5 | 0.2447 | 0.0016 | NA | Wood | -25.60 | 11,300 | 50 | 13,100 | 13,300 | 13,200 |
| OS-106863 | 524.6 | 0.2428 | 0.0011 | NA | Twig | -25.30 | 11,350 | 35 | 13,150 | 13,300 | 13,250 |
| OS-133660 | 541.6 | 0.2220 | 0.0015 | NA | Twig | -25.70 | 12,100 | 55 | 13,800 | 14,100 | 13,950 |
| OS-107085 | 567.6 | 0.2048 | 0.0014 | NA | Grass | -35.40 | 12,750 | 55 | 15,000 | 15,400 | 15,200 |

[a] Samples not used in the discussion due to possible hardwater effect

NA: Not Available.

NA δ13C: Sample was either too small or the measurement was not requested.

NA Mass: Not recorded





### 4.3 Optically stimulated luminescence dating

Our small-aliquot $D_e$ results from both 21SICK-01 and -02 show evidence of partial bleaching, as expected in a glaciofluvial environment (Table 3; AFig. 1 & 2; Rittenour et al., 2015). $D_e$ results from the two samples are considerably scattered, positively skewed, and have overdispersion values between ~30 and ~60%, all indicative of incomplete bleaching and justify the use of the MAM (e.g., Olley et al. (1999). Our two OSL MAM ages are 19.8 ± 2.6 and 20.6 ± 2.9 ka. The two samples are from within 10 cm of each other and yield statistically indistinguishable ages.

**Table 3. Optically Stimulated Luminescence Age Information**

| Sample num. | USU num. | Depth (m) | Num. of Analyses[1] | Dose Rate (Gy/kyr) | Equivalent Dose[2] ± 2σ (Gy) | OSL Age ± 1σ (ka) |
|---|---|---|---|---|---|---|
| 21-SICK-1 | USU-3622 | 2.05 | 21 (42) | 2.70 ± 0.11 | 53.55 ± 11.51 | **19.82 ± 2.60** |
| 21-SICK-2 | USU-3623 | 2.15 | 23 (37) | 2.23 ± 0.09 | 46.09 ± 10.07 | **20.63 ± 2.91** |

[1] Age analysis using the single-aliquot regenerative-dose procedure of Murray and Wintle (2000) on 0.4-1-mm small-aliquots (SA) of quartz sand (150-250 μm). Number of aliquots used in age calculation and number of aliquots in parentheses.

[2] Equivalent dose ($D_E$) calculated using the Minimum Age Model (MAM) of Galbraith and Roberts (2012).

## 5 Discussion

### 5.1 Stratigraphy

We interpret Unit 1 as the primary till that comprises the Kent Moraine. At the Vincent-1 site we cored from 4.1 to 6.6 m below the wetland surface (2.5 m), but only recovered 1.2 m due to compaction with the GeoProbe system. We assume we reached below the post-glacial infill and into the primary glacial deposit since this unit spans 2.5 m and we found no changes in stratigraphy (Fig. 5).

Given the hummocky nature of the moraines (Fig. 3) and the complex stratigraphy within Unit 2 (Fig. 5), we interpret this unit to record the transition from an ice-cored moraine to the modern kettled topography for both moraines. The most striking feature of the sediment cores are the numerous transitions between fine- and coarse-grained deposition. We interpret Unit 2 silt settled out of suspension in lacustrine conditions, indicating that all seven basins likely held small kettle lakes of shifting dimension during this period. We propose that the alternating clay and diamicton sediments captured in 20VIN4 are slumps of primary till into the kettle lake with otherwise clay-rich sedimentation; these slumps probably occurred as buried glacial ice melted and destabilized the basin's slopes. The stratigraphy from 21LPB1 is likely the result of the changing depositional environments on the moraine as the kettle formed. Higher energy deposits of sand and gravel at the base of the unit were likely deposited atop the ice-cored moraine in fluvial or shallow water settings before being redeposited, in stratigraphic position, by the melting of buried ice beneath them. These sediments then floored the new kettle lake and deposition of lacustrine silt began.



The transition in sediment type between Units 2 and 3 likely reflects a shift to a more productive lake and
landscape, in concert with increased stabilization of the surrounding moraine. Some layers of minerogenic sediment
in the bottom of Unit 3 in 20VIN1 & 20VIN3 show that the landscape continued to receive sediment from primary
glacial deposits after the transition to more organic-rich deposition. We infer that the inclusions of gray clay and
brown silty macrofossils in 20VIN1 are rip-up clasts by their clast-like appearance and stark contrast to the
surrounding sediment (Fig 5; Panel C). They were potentially frozen during the time of deposition. This further
suggests the presence of reworked material near the Unit 2/3 transition. The subsequent transition from lacustrine
organic-rich silt to peat (Lower and Upper Unit 3, respectively) records the shift from lake to bog/wetland due to the
filling of the basin, shallowing of the lake, and encroachment of the shoreline.

### 316 5.2 Chronology

The OSL ages support our estimated age of 25 – 20 ka for the Kent Moraine from prior literature and
affirms our confidence in the age assignments using correlations of dated features elsewhere. The OSL samples are
from 2 m below the surface of the ~70 m thick kame delta. The sample location within the topset beds of a
short-lived ice-contact delta suggests that our OSL samples constrain the time just before the ice sheet retreated and
ceased building the delta 19.8 ± 2.6 – 20.6 ± 2.9 ka.
The basal ages, taken at face value, indicate the deposition of the Kent Moraine occurred shortly before ~15
ka; this does not agree with our OSL age or the regional correlations. Furthermore, a Kent Moraine age of ~15 ka
contradicts the ~17 ka age for the Lake Escarpment Moraine, which lies up ice flow from the Kent Moraine. The
above information, combined with our evidence for an unstable landscape depicted from our sediment core
stratigraphy and numerous age reversals, suggests that our radiocarbon ages from Unit 2 consist of organic material
that was reworked into these kettles during kettle formation.
We have identified spores and seeds of aquatic plants *Chara* and *Potamogeton* (O. Bennike, personal
communication) among the macrofossils from samples 16,050-16,300 and 15,650-15,900 cal yr BP from 20VIN1
and sample 15,350-16,650 cal yr BP from 20VIN3. These samples also have enriched $\delta^{13}C$ values, suggesting that
these samples contained aquatic material (Deuser and Degens, 1967; Oana and Deevey, 1960; Wang and Wooller,
2006). Our sites lie within calcareous tills that overlie sedimentary bedrock (LaFleur, 1979; MacClintock and Apfel,
1944), which can add aged carbon to the lake water. Aquatic plants derive their carbon from lake water, so
radiocarbon ages from aquatic plants could produce radiocarbon ages that overestimate the age of the sample (the
'hardwater effect'; Deevey et al., 1954; Keeley and Sandquist, 1992).
We move forward using samples assumed to be terrestrial from visual identification and supported by $\delta^{13}C$
values. Four ages from 20VIN1 Unit 2 remain: 19,350-19,600, 15,050-15,550, 14,300-15,050, and 14,050-14,850
cal yr BP. We limit the 20VIN3 chronology to one trustworthy age of 14,350-15,150 cal yr BP. We derived the age
of 16,650-17,350 cal yr BP in 21LPB1 from a fish bone; a fish could be susceptible to the same hardwater effect as
aquatic vegetation, and thus we do not use it in our evaluation. Instead, we use the next lowest age of 13,600-14,000
cal yr BP as the basal age, along with 15,000-15,400 cal yr BP from 13DFK1.



These age estimates that are assumed to be more trustworthy still exhibit age reversals. These ages support
our interpretation from the visual stratigraphy that reworked sediments contain organic matter that does not
accurately date to the sediment's deposition. The macrofossil-rich rip-up clast in 20VIN1 holds evidence for two
important interpretations: 1) the landscape was ice-free and at least sparsely vegetated as early as 19,350-19,600 cal
yr BP (consistent with our OSL ages suggesting ice sheet retreat by $19.8 \pm 2.6 – 20.6 \pm 2.9$ ka), and 2) the landscape
stored this long-dead vegetation for thousands of years before it was redeposited.
Since we do not trust that radiocarbon ages from Unit 2 accurately date the time of sediment deposition,
and the moraine ages are incompatible with regional correlations, we do not interpret our lowest, basal ages to
record the timing of ice recession and abandonment of the moraines. Instead, we interpret these younger than
expected ages to record kettle basin formation and moraine stabilization for both the Kent and Lake Escarpment
moraines between 15,000-15,4000 and 13,600-14,000 cal yr BP. This interpretation also reconciles the similar basal
ages between both moraines that are likely several thousand years different in age.

**5.3 A model for kettle basin formation**

We propose the following post-glacial history in western New York (Fig. 6). The deposition of the Kent
Moraine occurred at least $19.8 \pm 2.6 – 20.6 \pm 2.9$ ka and the landform remained ice-cored for the ensuing $5 – 6$ kyr.
The deposition of the Lake Escarpment Moraine took place around 17 ka and likewise remained ice-cored for the
next $2 – 3$ kyr. The hummocky nature of the moraines indicate that they were ice-cored, and we suggest that
persistent buried glacial ice prohibited stabilization until well after deposition. Our interpretation is that after ~15 ka
buried ice began to melt, and morainal topography – including kettle basins – began to evolve more rapidly (Fig. 6
& 7). During the earliest stages of kettle basin formation, there was increased mobilization of sediments from within
the uneven ice-rich topography. These initial sediments contained both re-worked and contemporary organic matter
from the catchment and were deposited in our study sites as Unit 2. According to this interpretation, our radiocarbon
ages do not record the initial deposition of these moraines, but instead its stabilization ~15 to 14 ka.
Ice-cored moraines can remain ice-cored for thousands of years after deposition due to sediment cover that
insulates and preserves the buried ice (Florin and Wright, 1969). If the region is cold enough to support permafrost it
may extend the duration that the moraine remains ice-cored (Clayton et al., 2001; Henriksen et al., 2003;
Schomacker, 2008). Given that the kettles appear to have formed within ~1 kyr of each other, and their formation
coincided with the warm Bølling/Allerød period, this suggests the climate during Heinrich Stadial 1 may have been
cold enough to help preserve the ice.



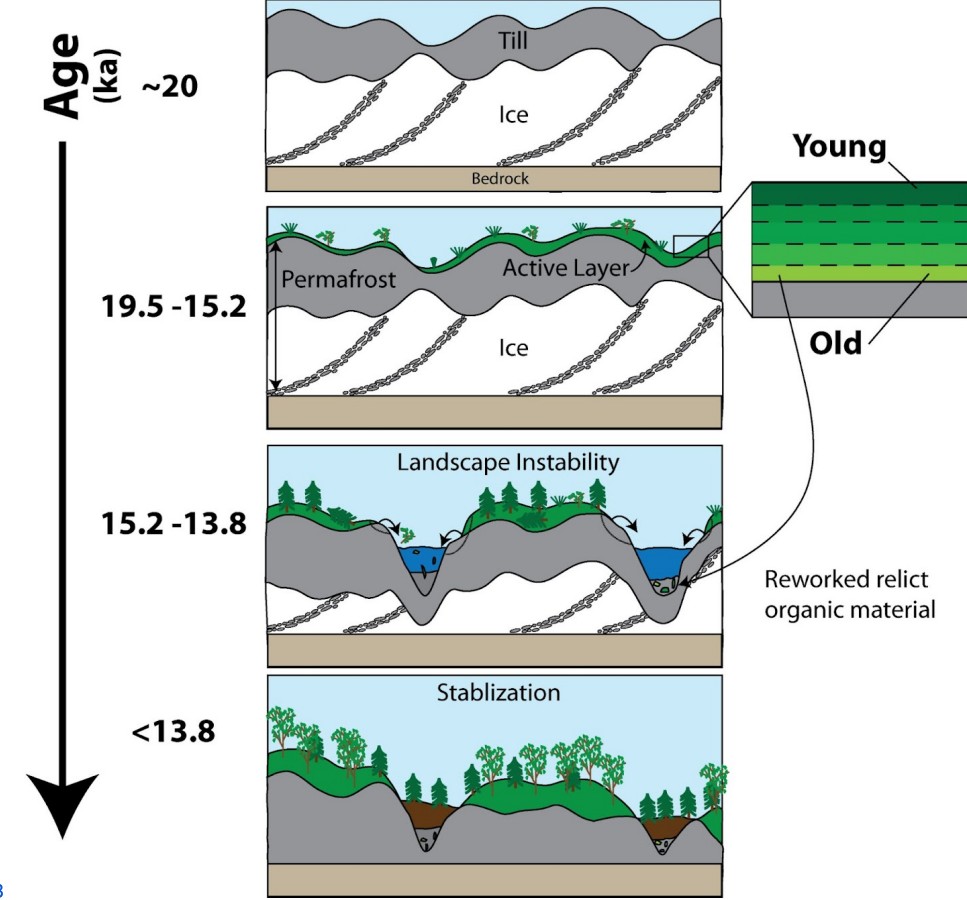


**Figure 6. Conceptual model of kettle basin formation of the Kent Moraine in western New York building on Florin and Wright (1969). The same model applies to the Lake Escarpment moraine, except the timeline begins ~17 ka. First, the LIS deposited the ice-cored Kent Moraine. It remained ice-cored, perhaps influenced by permafrost, while tundra vegetation grew atop the moraine and stored carbon in the soil. Next, during climate amelioration in the Bølling-Allerød periods, the ice in the moraine melted. This led to the formation of basins that filled with both contemporaneous and reworked sediments. This is also likely the time when trees and other organic material could be slumped and formed deposits that placed primary tills adjacent to younger material. Finally, organic-rich sediment deposition dominates after ~13.8 ka.**

## 5.4 Implications for the climate in western New York

The climate of western New York between 20 and 15 ka is poorly known, but records from Ontario, Ohio, and New England suggest the climate events of the North Atlantic influenced the northeastern U.S. These terrestrial climate reconstructions depict a cold Heinrich Stadial 1 (~18 to ~14.7 ka), a shift to warmer temperatures during the Bølling-Allerød, and a cool Younger Dryas (Gill et al., 2012; Gonzales and Grimm, 2009; Grigg et al., 2021; Shuman et al., 2002; Watson et al., 2018; Yu, 2007; Yu and Eicher, 1998). A stable Heinrich Stadial 1 and shift to





warmer temperatures during the Bølling-Allerød is shown by Watson et al. (2018), who used biomarkers
(branched-GDGTs) to report that mean annual temperature in central Ohio varied between –2.0 and -0.5 °C from
17.0 to 14.5 ka before warming 5°C between 14.5 and 13.0 ka.
The rate of LIS retreat offers additional insight into the climate in the northeast US. Barth et al. (2019) used
cosmogenic nuclide dating of glacially-transported boulders to estimate LIS thinning in the Adirondack Mountains
and showed increased thinning between 15.4 ± 1.0 and 13.9 ± 0.9 ka, generally coincident with the Bølling. The
New England Varve Chronology shows a relatively steady net retreat rate of the LIS through the Hudson Valley
between 18 and 14.7 ka; during the Bølling the net retreat rate tripled, implying that New England experienced
elevated warmth at that time (Ridge et al., 2012).
Ice-wedge casts can be used to identify areas that experienced past permafrost and constrain past
temperature because their formation requires mean annual temperatures between -6 to -8°C (French, 2007; French
and Miller, 2014). Ice-wedge casts are preserved in southern Ontario that were deposited 18-15 ka based on regional
correlations (Dalton et al., 2020; Gao, 2005; Morgan et al., 1982). This suggests that the mean annual air
temperature was low enough near our study site during Heinrich Stadial 1 to support permafrost. While this
temperature depression is larger than reported by Watson et al. (2018), it's likely there was a strong temperature
gradient between Ohio and western New York during deglaciation, with the latter remaining within 100 km of the
ice margin until 14 ka (Dalton et al., 2020). This proximity to the ice sheet from the LGM to 14 ka may have been a
driver of the cold climate that persisted in western New York. There are no reports of relict permafrost features
within the LGM limit in western New York, but their presence south of the LGM extent suggest the likelihood of
permafrost within the limit as well (French and Millar, 2014).
Finally, there are seven local pollen records from Miller (1973), Calkin and McAndrews (1980), and Doody
(2018) that describe the initial deglacial vegetation in western New York. Only the Allenberg Bog (Miller, 1973) and
Dragonfly Kettle (Doody, 2018) pollen records captured a 'tundra' zone at the base, although the presence of both
arctic and temperate vegetation complicates their interpretation. Given our results, we believe this pollen tundra
zone captured both the tundra vegetation that was growing on the moraine prior to basin formation and the more
temperate vegetation as spruce and pine moved in during the Bølling. Unfortunately, the pollen records may be
unreliable before 14 ka due to the same reworking problems as our radiocarbon dating, but this remains site specific.
The tundra zone is overlain by an interval with high spruce and pine pollen; this is the lowest unit found in the other
five records (Miller, 1973; Calkin and McAndrews, 1980). This is likely reflecting the new forest biome associated
with warmer temperatures.
Altogether, there is evidence that the lag time between ice sheet retreat and kettle basin stabilization may be
attributable to sustained permafrost in western New York due to cold North Atlantic conditions during Heinrich
Stadial 1 (Fig. 7). The warming at the Bølling onset at ~14.7 ka may have increased regional temperatures, causing
the melting of buried ice, initiating a phase of rapid landscape evolution and the formation of kettle basins, and
eventually stabilizing the morainal topography. Numerous studies discuss the role of permafrost in the lag time
between moraine ages and basal macrofossils along the south-central LIS margin, including Indiana and Illinois
(Curry et al., 2018; Fisher et al., 2020), Michigan (Yansa et al., 2020), and Wisconsin (Clayton et al., 2008).



Our findings support the observations and conclusions from numerous studies that radiocarbon dates can be

extreme minimum age constraints on deglaciation (Curry et al., 2018; Fisher et al., 2020; Florin and Wright, 1969;
Halsted et al., 2023; Yansa et al., 2020). In New England, minimum-limiting radiocarbon ages may be the reason for
the discrepancy between the timing of moraine deposition as recorded by $^{10}$Be exposure dating (e.g., Balco et al.,
2002; Corbett et al., 2017) and radiocarbon ages of basal macrofossils in lakes and bogs (e.g., Peteet et al., 2012).
The younger than expected radiocarbon ages from the Valley Heads Moraine from Kozlowski et al. (2018) may be
afflicted by similar processes. Permafrost during Heinrich Stadial 1 may have minimized landscape evolution in
New England and central New York as well and could help explain the offset.

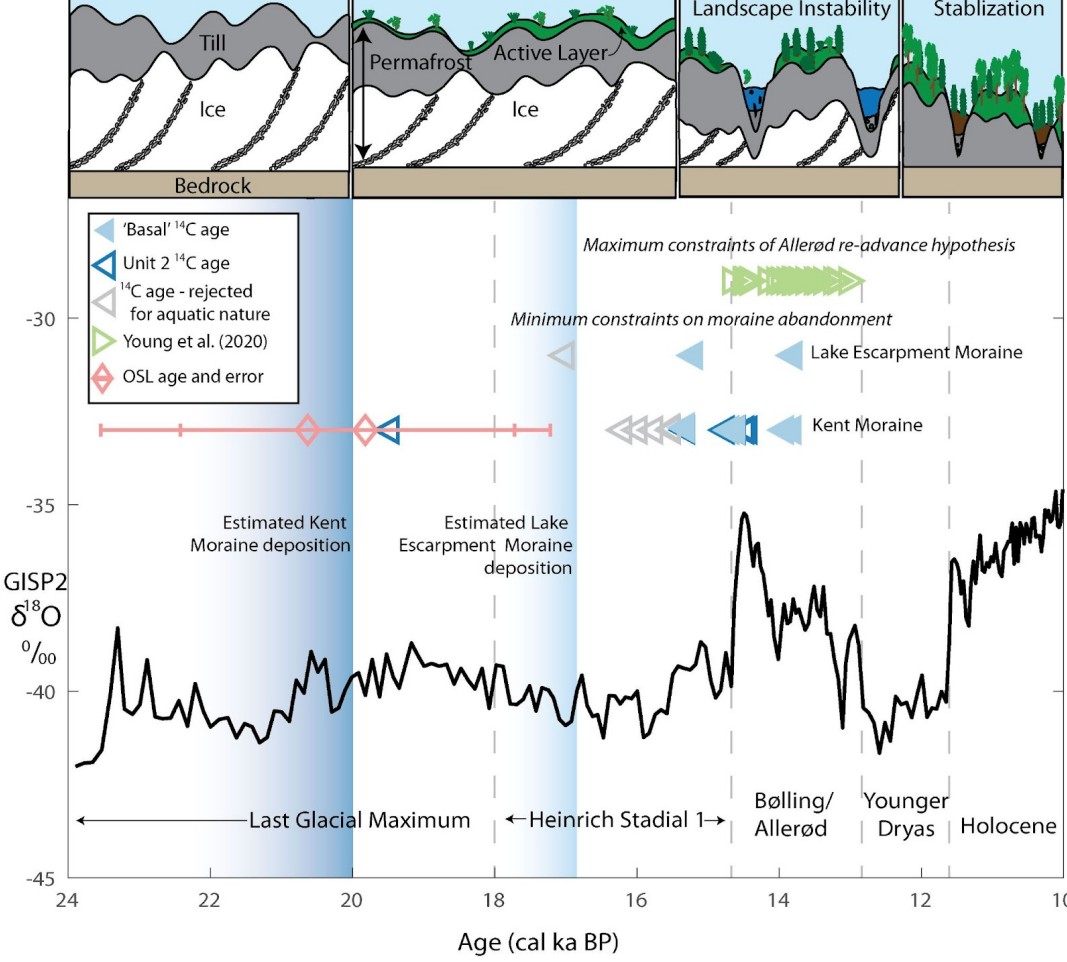


**Figure 7. Comparison of radiocarbon ages from the Kent and Lake Escarpment moraine and Young et al. (2020) in the**
**context of North Atlantic deglacial climate changes. Black line is the GISP2 δ¹⁸O record (Grootes and Stuiver, 1999). Dark**
**blue and light blue fading is the estimated deposition of the Kent Moraine and Lake Escarpment Moraine, respectively.**
**Dark blue and light blue triangles are the lowest reliable radiocarbon ages from the Kent and Lake Escarpment Moraine**
**sediment cores, respectively. Gray triangles are radiocarbon ages that we suspect have hardwater contamination. Pink**
**diamonds are OSL ages and 2σ errors from the kame delta outboard the Kent Moraine. Green triangles are ages from**



**Young et al. (2020) interpreted by them to be maximum-limiting constraints on the 13 ka re-advance. Errors for all**
**radiocarbon dates are not plotted because their width is smaller than the symbols.**

**5.5 Allerød re-advance hypothesis**

444   The stratigraphically lowest radiocarbon ages from Unit 3 in the Lake Escarpment Moraine kettle basins,
which are 15,000-15,400 and 13,600-14,000 cal yr BP, pre-date the ~13.1 ka re-advance suggested by Young et al.
(2020) (Fig. 5 & 7). Chronologically constrained organic-rich sedimentation, with no stratigraphic evidence of
interruption, ensued from at least 13,600-14,000 cal yr BP and well into the Holocene. Furthermore, there is no
evidence of over-compaction in our bulk density measurements in 21LPB1 during this interval of time (Fig. 5).
Thus, we do not find evidence that a ~13.1 ka LIS advance created or overran the Lake Escarpment Moraine as
hypothesized by Young et al. (2020). Rather, we suggest that the landscape was unstable during its transition from a
permafrost-dominated landscape to one with evolving and then stabilizing morainal topography. This landscape
instability with reworking of glacial sediments may have led to the stratigraphy interpreted by Young et al. (2020) as
primary tills in contact with trees dating to 13 ka (Fig. 7). Both the Dragonfly and Little Protection sites have
intervals with increased wood deposition between 14 and 13 ka and future work could investigate the source of these
woody intervals to further investigate the results from Young et al. (2020).

**6 Conclusion**

458   We present 41 new macrofossil-based radiocarbon ages from kettle basin infills in western New York. We
find that the lowest reliable radiocarbon ages between 15,000-15,400 and 13,600-14,000 cal yr BP are 2 – 6 kyr
younger than our OSL age constraints on moraine deposition of 19.8 ± 2.6 – 20.6 ± 2.9 ka. We interpret this offset to
be due to a cold climate in western New York during Heinrich Stadial 1 supporting persistent buried ice which
inhibited kettle basin formation until regional warming that took place during the Bølling. Our results do not support
a re-advance of the LIS over the Lake Escarpment Moraine ~13 ka (c.f. Young et al., 2020). The lag time between
ice sheet retreat and moraine stabilization in western New York may present an alternate explanation for
inconsistencies between basal ages in sediment cores and other dating methods in central New York (Kozlowski et
al., 2018) and eastern New York (Peteet et al., 2012).

467   Future work could target features that are stable during ice retreat even where permafrost is present, such as
outcrops of pro-glacial and ice-walled lake plane deposits (e.g., Curry et al., 2018), or perhaps moraines that are not
hummocky in nature. This limitation may not be as necessary in environments where climate more quickly
ameliorated, such as appears to have been the case in southern Ohio (Glover et al., 2011). Additionally, it may be
important to consider the coring equipment. The GeoProbe coring device enabled us to collect stiff mineral-rich
sediments lower than otherwise possible with the Livingstone and Russian Peat coring devices. This meant that our
coring did not stop at first contact with stiff minerogenic sediment that could mistakenly be interpreted as primary
glacial in origin.



**Appendix A.**

### Table A1: Dose Rate Information

| USU num. | Lat/Long | In-situ $H_2O$ (%) | $D_R$ Subsample[1] | K (%)[2] | Rb (ppm)[2] | Th (ppm)[2] | U (ppm)[2] | Cosmic (Gy/kyr) |
|---|---|---|---|---|---|---|---|---|
| USU-3622 | 42.11394/ -78.94899 | 7.5 | F: 70% | 1.64±0.04 | 77.6±3.1 | 7.8±0.7 | 2.6±0.2 | 0.18±0.02 |
| | | | M: 20% | 1.12±0.03 | 58.7±2.3 | 8.6±0.8 | 2.2±0.2 | |
| | | | C: 10% | 1.29±0.03 | 76.1±3.0 | 11.1±1.0 | 2.1±0.1 | |
| USU-3623 | 42.11394/ -78.94899 | 20.0 | F: 85% | 1.52±0.04 | 74.4±3.0 | 8.3±0.7 | 2.0±0.1 | 0.18±0.02 |
| | | | M: 15% | 1.35±0.03 | 72.7±2.9 | 8.4±0.8 | 2.4±0.2 | |

[1] Dose rate ($D_R$) subsamples based on grain size: fine-F (<1.7 mm), medium-M (1.7-16 mm), coarse-C (>16 mm), and weighted proportions (%) of subsamples used with chemistry in gamma dose rate calculation. Beta dose rate uses chemistry from fine fraction (<1.7 mm) only.

[2] Radioelemental concentrations determined using ICP-MS and ICP-AES techniques; dose rate is derived from concentrations by conversion factors from Guérin et al. (2011).


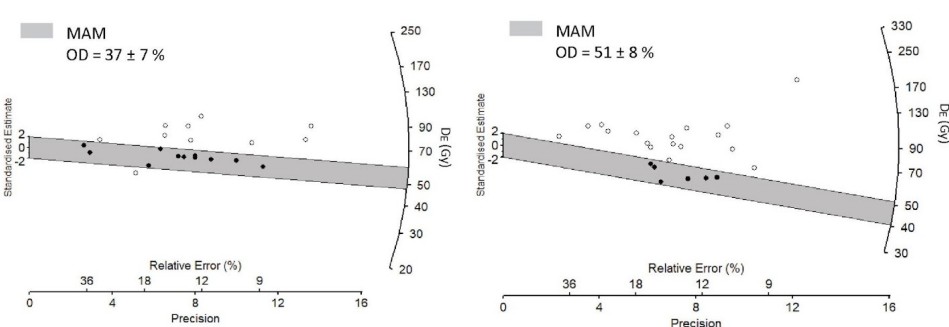


**Figure A1. Equivalent dose ($D_E$) distributions for the luminescence samples collected from the kame delta associated with**
**an ice-margin position near the Kent moraine. MAM = minimum age model of Galbraith and Roberts (2012) fit to the $D_E$**
**data (gray shaded region). OD = overdispersion, a metric of $D_E$ scatter beyond instrumental error, where OD > 30% is**
**interpreted to be due to partial bleaching due to incomplete solar resetting of the luminescence signals in the quartz**
**grains.**



USU-3622

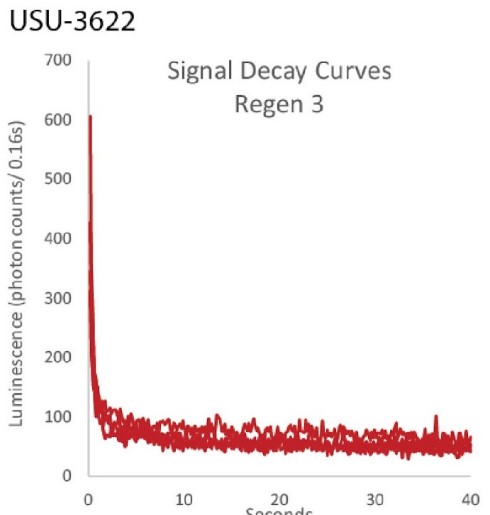

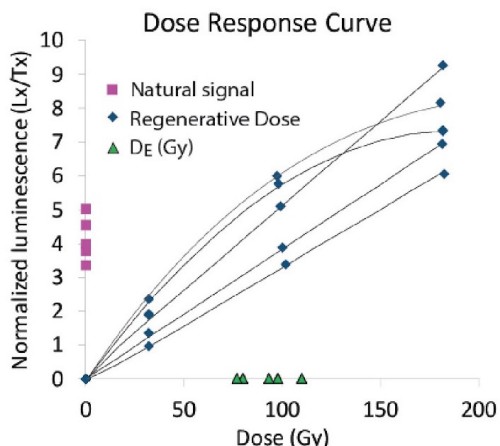

USU-3623

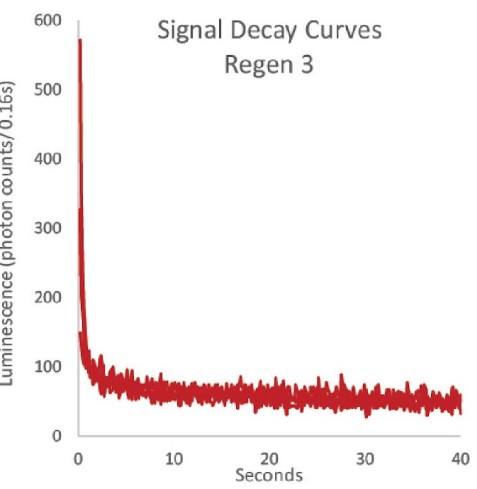

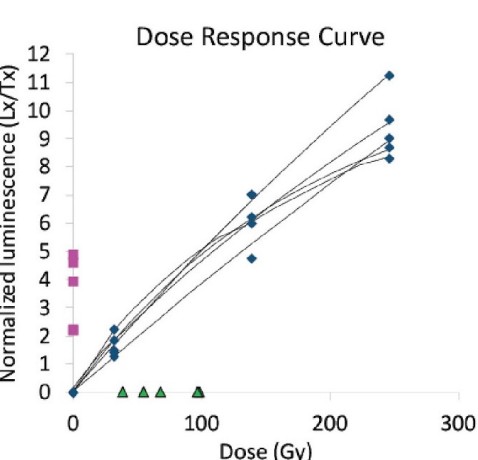

**Figure A2. Example luminescence signal decay (left) and dose-response curves from 5 aliquots from each of the luminescence samples.**

**Data availability:** Table 2 provides the data to calculate the radiocarbon ages from this study. Table 3 and Table A1 provide the data to calculate OSL ages from this study.

**Author contributions:** JPB and KKP conceptualized the study. ALK and KKP provided funding for fieldwork and lab analyses. KKP, JPB, CKW, BMC, and EPY collected sediment cores. KKP, BMC, EPY and JPB conducted



downcore analyses and radiocarbon sampling. CKW collected OSL samples and TMR conducted lab analyses and
calculated the ages. KKP compiled and recalculated the radiocarbon ages. KKP, JPB, ALK, CKW, and BMC
interpreted the results. KKP wrote the first draft of the manuscript and all authors contributed to editing. KKP and
TMR developed the figures and tables.

**Competing interests.** The authors declare that they have no conflict of interest.

**Acknowledgements:** We thank the Vincent, Songster, Gebhard, and Bohall families for the access to their property,
as well as their enthusiasm and friendship. We thank Joseph Tulenko, Brandon Graham, Elizabeth Thomas, Kurt
Lindberg, Owen Cowling, Fiona Ellsworth, Joshua Charlton, Liza Wilson, Jason Parsons, Will Phillips, and George
Thomas for their help in the field (it takes a village!). We thank the National Ocean Sciences Accelerator Mass
Spectrometry and W. M. Keck Carbon Cycle Accelerator Mass Spectrometer laboratories for radiocarbon analyses.
We thank the Luminescence Lab at Utah State University for OSL analyses.

**Funding sources:** This research was supported by the United State Geological Survey Great Lakes Geological
Mapping Coalition grant #G20AC00418, the NSF/GSA Graduate Student Geoscience grant # 13056-21, which was
funded by NSF Award # 1949901, and the Mark Diamond Research Fund of the Graduate Student Association of
the State University of New York at Buffalo.

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
