# Peer review of "New age constraints reveal moraine stabilization thousands of"

_EGUsphere, 2023_

## Referee Comment (RC2)

**Prince et al. (2024) – Peer review**
*Geochronology*

The submitted manuscript by Prince et al. presents new geochronologic data in the form of basal radiocarbon ages from kettle lakes and optically-stimulated luminescence (OSL) ages from glacial deltaic deposits to constrain the timing of the Kent and Lake Escarpment moraines – a Last Glacial Maximum (LGM) and recessional moraine, respectively. Results from these data are used to refine the timing of Laurentide Ice Sheet (LIS) deglaciation in western New York (NY) and address the proposed Allerød re-advance hypothesis (Young et al., 2020) which proposes re-advance of the LIS ~13 ka beyond the Lake Escarpment moraine.

The authors proposed a landscape evolution of kettle formation thousands of years after retreat of the LIS from the respective margins leading to a lag response in radiocarbon ages relative to the onset of deglaciation. The mechanism for the observed lag is attributed to the persistence of ice-cored moraines and permafrost across the Heinrich Stadial 1 cold interval followed by melting of the ice features during the abrupt Bølling warming leading to kettle lake formation. Interbedded coarse sediments amongst silt-dominated lake deposits are interpreted as periodic slumps of till during moraine stabilization.

The authors find no evidence to support re-advance of the LIS across the Lake Escarpment moraine and propose their new interpretation of landscape/kettle stabilization to explain the alternating coarse-fine sediments within the cores.

Overall, Prince et al. present compelling evidence to support their interpretations and conclusions. The results of this manuscript will advance the community's understanding of western NY deglaciation and the mechanisms driving ice-sheet deglaciation. There is room for improvement within the manuscript itself including some organizational changes and clarification of certain arguments related to radiocarbon interpretation. However, should these issues be addressed by the authors it is my opinion that the submitted manuscript should be accepted for publication.

My comments for improvement are listed below as broad, sections-specific, and finer, line-specific comments.

*Section-specific comments:*

Section 3. There are no issues with the content of this section. However, many of the sentences start with the word "we" (e.g., "We collected…", "We determined…", "We returned…", etc.) often in sequential sentences. I recognize that there is debate within the scientific community about the use of active or passive voice in writing, but for now the Methods section would read more fluidly if many of these sentences were changed to "Samples collected were analyzed for…" or "OSL analysis was conducted at…".

Section 4.1. It's easy to get lost in the stratigraphic descriptions for each core in Section 4.1. It could be helpful for the reader to explicitly discuss each core based on the associated moraine, so it becomes easier to follow when the stratigraphy of a Lake Escarpment moraine core is being described versus a Kent moraine core.

Section 4.2. It could be helpful to declare the stratigraphic unit in which each basal age is found since that is not consistent across all cores. As an example, starting on line 264 the basal ages for the Kent moraine are described, yet the 15,050-15,550 cal yr BP ages from 20VIN1 are in Unit 2 just above the boundary with Unit 1, whereas the 13,00-14,050 cal yr BP ages from 15ABB7 are found in Unit 3. Clarifying this for all radiocarbon data in Section 4.2 will strengthen your argument and allow the reader to immediately

associate the ages with their stratigraphic unit, at the same time setting up the stratigraphic discordance with older ages higher up the core.

Section 5.2. The second paragraph of Section 5.2 discusses the basal radiocarbon ages in relation to the timing of LIS retreat from the Kent moraine. It would be helpful to clarify you are referring to the ages from 20VIN1 located in Unit 2 just above Unit 1 which is interpreted as the glacial till. The description of "shortly before ~15 ka" needs to be changed to "prior to ~15 ka" as these are minimum-limiting ages on deglaciation and the duration between deglacial onset, as indicated as the stratigraphic change from Unit 1 to Unit 2, and the radiocarbon ages cannot be determined from these data alone. As such, there is not necessarily a contradiction between the young basal radiocarbon ages and regional correlations with the Lake Escarpment moraine ~17 ka. These results simply state that abandonment of the Kent moraine happened prior to ~15 ka and must have occurred prior to deposition of the Lake Escarpment moraine ~17 ka. Additionally, this interpretation of the basal radiocarbon age lends support to the eventual age-lag conclusion due to persistent ice and permafrost within the moraine.

*Line specific comments:*

Line 36: Consider revising. "Well constrained ice sheet chronologies…constrain". Refine?

Line 93: Consider revising. "…dated to…by radiocarbon dating" is redundant.

Line 116: Define "significant" if you plan to describe the re-advance in this way.

Line 154: Was sediment bulk density only measured on the Little Protection cores? If so, why not on the others?

Line 283: Missing an end parenthesis after Olley et al. (1999).

Line 290: Consider using consistent terminology when discussing cores. Here you describe Vincent-1 when previously these cores were described as "20VIN1". You could also include the associated cores in parentheses after "Vincent-1" for clarity.

Line 300: Careful with the word "probably". This is an interpretation based on stratigraphy alone. A safer word to use here is "potentially".

Line 301: Reiterate that core 21LPB1 is associated with the Lake Escarpment moraine.

Line 330: Clarify that "These samples…" refers to the macrofossils.

Line 336: Greater description is needed for how samples were assumed to be terrestrial origin. What visual cues were looked for to identify terrestrial vs. aquatic samples.

Line 348: Consider revising. The basal radiocarbon ages are trustworthy, but the up-core ages exhibit stratigraphic discordance and therefore do not reflect an accurate age of sediment deposition.

Line 364: The radiocarbon ages are from sediments stratigraphically above the glacial deposits therefore would not reflect moraine deposition. Do you mean "do not record deglacial onset"? A more accurate conclusion is reached on Line 425 "…radiocarbon dates can be extreme minimum age constraints on deglaciation."

Line 366: Consider revising. "Ice cored moraines remained *as such…*" or "Moraines can remain ice cored for…"

*References*
Young, R. A., Gordon, L. M., Owen, L. A., Huot, S., and Zerfas, T. D.: Evidence for a late glacial advance near the beginning of the Younger Dryas in western New York State: An event postdating the record for local Laurentide ice sheet recession, Geosphere, https://doi.org/10.1130/GES02257.1 , 2020.

---

## Author Comment (AC1)

We appreciate the detailed and thorough reviews from Referee 1, 2 and 3. Between the three reviews, there are common themes, which we group together and respond to at once when appropriate. Below are responses to 1) their general comments, and 2) line-by-line comments. We also include our response to the editor's comment on our initial revision.

The reviewer comments are unbolded and marked as Referee 1 (*R1*), Referee 2 (*R2*), and Referee 3 (*R3*). The author's responses are marked as *AR* and **bolded**. This same document is included in all three replies and responds to all referee and editorial comments.

*Presentation and discussion of the Allerød Re-advance Hypothesis*

*Referee 1 (R1):* 2) The "controversial" hypothesis should be presented in more detail in the introduction or geological setting. Now it is only briefly mentioned in the introduction and again in the discussion.  It is very relevant to describe in detail how Young et al concluded that a readvance took place at 13 ka.

*Author Response (AR):* **We will update the introduction to include the following text in red:**

**"However, Young et al. (2020) recently interpreted new and existing radiocarbon ages from western New York to support a significant re-advance of the LIS at ~13 ka that overtopped the Lake Escarpment Moraine and nearly reached the Kent Moraine (Fig. 1). The evidence includes the re-interpretation of several unrelated sites throughout western New York, but largely hinges on new trenched sections near the Kent Moraine revealing logs in clayey diamicton, which Young et al. (2020) suggest requires glacial overriding of a forest ~13.3 to ~13.0 ka. In contrast to Young et al.'s (2020) reconstruction, most literature places the LIS margin north of Lake Ontario at this time (Dalton et al., 2020; Muller and Calkin, 1993; Terasmae, 1980; and references therein), with the drainage of Glacial Lake Iroquois occurring at ~13 ka (Fig. 1; Cronin et al., 2012; Lewis and Anderson, 2019; Rayburn et al., 2005). To reconcile the disagreement in timing between the hypothesized Allerød re-advance and existing chronologies, Young et al. (2020) invoke a largely floating ice mass that left minimal traces of its existence in most areas. If a re-advance of the scale hypothesized by Young et al. (2020) occurred (henceforth referred to as the 'Allerød re-advance hypothesis'), we would need to revisit many regional deglaciation chronologies.**

*Presentation, organization, and discussion of the stratigraphy:*

*R2:* Section 4.1. It's easy to get lost in the stratigraphic descriptions for each core in Section 4.1. It could be helpful for the reader to explicitly discuss each core based on the associated moraine, so it becomes easier to follow when the stratigraphy of a Lake Escarpment moraine core is being described versus a Kent moraine core.

*R3:* The *stratigraphy* and *sediment core chronology* sections need some deep reworking. Indeed, I found these sections poorly organized, making it difficult to read and understand the

results. The different cores are mixed in the text and the different units are not well defined. A better presentation of the data is important, as the position of the samples within the units is subsequently used extensively to correlate ages. Suggestions for improvement are included in my comments line by line in the "specific comments".

*AR:* **These two comments suggest a reformatting of the results section to group the stratigraphy and radiocarbon results by sediment core and moraine. We will format the results sections to discuss each site in order by moraine, describing each sediment core individually. This individual description will include 1) descriptions of the core stratigraphy by Unit, 2) radiocarbon ages with information about the unit they come from and 3) the age-depth relationship.**

*Presentation, organization, and discussion of the radiocarbon:*

*Associate editor:* Concerning the table 2 and reporting all calibrated intervals associated to the relative probability. I'll stick on the recommendations of the 14C community (e.g. Millard 2014). By reporting the interval that encompasses all calibrated intervals, you miss important information: the time periods that are unlikely and you no longer report a 95% confidence probability but something higher, between 0.95 and 1 since you include part of the remaining 5%. This is not mathematically correct. I would also draw your attention to the fact that the median makes no sense in the context of a multiinterval calibration. You may even end up with a median in a non-probable interval. You'll save space by eliminating this column. Furthermore, as the d13C are reported with uncertainty of 0.1‰, a single digit will sufficient to report this value. The spare room can be used to report 14C lab code. I understand your desire for an elegant table. My recommendation is to play with the thickness and color of the lines.

*AR:* **In Table 2 we will report the discrete solutions within the 95% confidence interval (as shown in your example table) instead of the min and max of this range (as it is reported now). We will truncate the d13C to one decimal point.**

*R1:* 1) The radiocarbon ages a given as ranges (min-max) throughout the text. Although this is the most correct way of reporting radiocarbon ages it makes the text less readable. I suggest that the min-max ranges are provided in the table and calibrated ages (in kiloyears) are used in the manuscript.

*AR:* **If the editor agrees, we would change the age presentation in the text and figures to median ages reported from Calib and uncertainties as the larger difference between the median and the maximum and minimum age, like this: "X.X ± X.X cal ka BP". Table 2 will still list ages in min-max form including discrete intervals as described above. Otherwise we will keep the age presentation in the text as the full 2-sigma range, as it is now, and refer to the table for more detailed information.**

*R1:* 3) It would be worthwhile to consider making age-depth models for the records where there are many radiocarbon ages. This would allow a better assessment of the potential outliers mentioned in the text and also plot the proxy data on an age scale.

*R3:* Radiocarbon ages should be placed in a chronological framework using chronostratigraphic models (in figure 5 or in the appendix) instead of just reported ages in a stratigraphic column as discussed in the main text. This should be easily done with Oxcal for example (using sequence or phase command…). Then we may have access to probability spectra, for individually calibrated and modelled ages. This would make it possible to better justify/approve the choices made by the authors to conclude that the ages are not those expected. This also allows them to better constrain the ages between the bottom and the top of the different cores. Maybe the radiocarbon ages will perhaps also be less rejected by the authors…

*AR:* **These two comments both suggest making age-depth models for the cores where we have sufficient radiocarbon constraints, so we will address their comments together. We agree that the age-depth plots provide a nice framework for discussing the radiocarbon results. We will add a supplemental file to our paper that contains age-depth plots so readers can visualize the sample distribution. We find the data shown as age-depth plots is most useful because we can symbolize the data by single terrestrial radiocarbon sample vs combined macrofossils with aquatic influence, etc., whereas an age-depth model created from rBacon or OxCal cannot. The 2σ age range is typically smaller than the symbol to show the sample age when viewing all the dates on a single plot, and the probability spectra for each age can be created with the raw data should a reader want to access this. Finally, our interpretation that Unit 2 records the collapse of ice-cored moraines and the creation of kettles in an unstable environment makes us hesitant to create an age-depth model through these sediments.**

*R2:* Section 4.2. It could be helpful to declare the stratigraphic unit in which each basal age is found since that is not consistent across all cores. As an example, starting on line 264 the basal ages for the Kent moraine are described, yet the 15,050-15,550 cal yr BP ages from 20VIN1 are in Unit 2 just above the boundary with Unit 1, whereas the 13,00-14,050 cal yr BP ages from 15ABB7 are found in Unit 3. Clarifying this for all radiocarbon data in Section 4.2 will strengthen your argument and allow the reader to immediately associate the ages with their stratigraphic unit, at the same time setting up the stratigraphic discordance with older ages higher up the core.

*AR:* **This will be clarified in the new structure of the results section.**

*R2:* Section 5.2. The second paragraph of Section 5.2 discusses the basal radiocarbon ages in relation to the timing of LIS retreat from the Kent moraine. It would be helpful to clarify you are referring to the ages from 20VIN1 located in Unit 2 just above Unit 1 which is interpreted as the glacial till. The description of "shortly before ~15 ka" needs to be changed to "prior to ~15 ka" as these are minimum-limiting ages on deglaciation and the duration between deglacial onset, as

indicated as the stratigraphic change from Unit 1 to Unit 2, and the radiocarbon ages cannot be determined from these data alone. As such, there is not necessarily a contradiction between the young basal radiocarbon ages and regional correlations with the Lake Escarpment moraine ~17 ka. These results simply state that abandonment of the Kent moraine happened prior to ~15 ka and must have occurred prior to deposition of the Lake Escarpment moraine ~17 ka. Additionally, this interpretation of the basal radiocarbon age lends support to the eventual age-lag conclusion due to persistent ice and permafrost within the moraine.

**AR: We agree that the nature of minimum-limiting radiocarbon ages means the ages are not contradictory to any of the correlations or the OSL ages. We will clarify wording according to this suggestion. A couple of line-by-line comments from R3 also suggest clarifying our discussion section that we address in more detail below.**

*Methods section:*

*R2:* Section 3. There are no issues with the content of this section. However, many of the sentences start with the word "we" (e.g., "We collected…", "We determined…", "We returned…", etc.) often in sequential sentences.  I recognize that there is debate within the scientific community about the use of active or passive voice in writing, but for now the Methods section would read more fluidly if many of these sentences were changed to "Samples collected were analyzed for…" or "OSL analysis was conducted at…".

**AR: We appreciate this comment aimed at streamlining our writing. That said, this comment is a bit subjective (first vs. third person a matter of writers' preference) and would like this in active voice.**

*Line-by-line comments:*

**R1:**

Line 15: consider using another word than "provocative" - it gives the wrong impression.  Maybe use an alternative instead of provocative.

**AR: We agree to change the adverb provocative, perhaps to 'controversial', as described above.**

Line 19: ..luminescence ages..."of what?

**AR: We will include that these ages are from topset beds in an ice-contact delta.**

Line 29: see the first comment about "provocative"

**AR: See above reply.**

Line 50: More information is needed about how Young interpreted the radiocarbon ages (see general comment)

*AR:* **See above reply.**

Line 69 (Fig. 1): Nice figure. Maybe add the existing chronological constraints mentioned in the text.

*AR:* **We will add in data discussed in the text as points along the moraines.**

Line 173 (Fig 3): Nice figure but consider changing the yellow colour or increasing the line thickness.

*AR:* **We will increase the line thickness.**

Line 214 (Fig. 4): Overall good figure but panel A could be improved. The dimensions seem off and I wonder what the light brown colour below (a) represents.

*AR:* **R3 also has comments on how to improve this figure. We will sub-out the image in Figure 4, check the alignments and dimensions, and provide in the figure that the light brown is also bedrock.**

Line 224: Ok descriptions, but it would be good to include the proxy data more in the description of the 3 units. In particular, MS, CaCO3 and water content could be better incorporated in the text.

*AR:* **We believe these data are best shown in the figures – we will refer to the graphs in the beginning of the results sections to guide readers.**

Line 249 (Fig. 5): Good summary figure with proxy data from the sediment cores. In 15ABB7 MS is 0 - is that a mistake? Also, some LOI and water content data are missing in 20VIN4.

*AR***: Yes ABB7 has MS values of zero for the entire core. We did not measure LOI and water content in 20VIN4 because it was a diamicton and difficult to sample, and we will add that information into the figure.**

Line 317-321: Consider starting with summarizing new data before stating it supports the existing data.

*AR:* **This was also mentioned by R3 – we will move this sentence to the end of the paragraph.**

**R2:**

*Line specific comments:*

Line 36: Consider revising. "Well constrained ice sheet chronologies…constrain". Refine?

*AR:* **We would change the wording to: '…are necessary to *determine* the timing of …".**

Line 93: Consider revising. "…dated to…by radiocarbon dating" is redundant.

*AR:* **We would change the wording to: '…basin *around* 17 – 16 cal ka BP *based on* radiocarbon dating…'.**

Line 116: Define "significant" if you plan to describe the re-advance in this way.

*AR:* **Similar to R1 comments about 'provocative', we will revise.**

Line 154: Was sediment bulk density only measured on the Little Protection cores? If so, why not on the others?

*AR:* **The data are only from Little Protection because we investigated the Allerød re-advance in our two cores from the Lake Escarpment Moraine, and Dragonfly Kettle data creation took place before the Allerød re-advance hypothesis was published and we did not measure bulk density. We will include this information on Line 155.**

Line 283: Missing an end parenthesis after Olley et al. (1999).

*AR:* **Thanks! We will correct that.**

Line 290: Consider using consistent terminology when discussing cores. Here you describe Vincent-1 when previously these cores were described as "20VIN1". You could also include the associated cores in parentheses after "Vincent-1" for clarity.

*AR:* **We will include the core name in parentheses after Vincent-1 for clarity. We will do the same if there is another occurrence of this.**

Line 300: Careful with the word "probably". This is an interpretation based on stratigraphy alone. A safer word to use here is "potentially".

*AR:* **We will change this to potentially.**

Line 301: Reiterate that core 21LPB1 is associated with the Lake Escarpment moraine.

**AR: We will clarify which moraine the core is from within this section.**

Line 330: Clarify that "These samples…" refers to the macrofossils.

**AR: Thanks, we will clarify that.**

Line 336: Greater description is needed for how samples were assumed to be terrestrial origin. What visual cues were looked for to identify terrestrial vs. aquatic samples.

*AR:* **Identification was rare at the time of sampling, partly due to the small size of the macrofossils available to be collected. Dr. Ole Bennike identified some of the dated samples. We opted for measurements of $\delta^{13}C$ to provide a basis to infer terrestrial vs aquatic nature of samples dated. The samples that we identified as likely aquatic material had identifiable spores of aquatic material and the samples inferred to be terrestrial do not. So, we will change this sentence to be 'We move forward using samples assumed to be terrestrial from a lack of identifiable aquatic macrofossils and supported by $\delta^{13}C$ values'.**

Line 348: Consider revising. The basal radiocarbon ages are trustworthy, but the up-core ages exhibit stratigraphic discordance and therefore do not reflect an accurate age of sediment deposition.

*AR:* **The Unit 2 ages are trustworthy as minimum-limiting constraints on moraine abandonment, but the evidence for slumps and rip-up clasts in Unit 2, plus the stratigraphic discordance in radiocarbon ages, are reasons to doubt the reliability of radiocarbon ages to reflect the age of the sediment they are within. We will include these reasons within this paragraph (Line 348) to clarify.**

Line 364: The radiocarbon ages are from sediments stratigraphically above the glacial deposits therefore would not reflect moraine deposition. Do you mean "do not record deglacial onset"? A more accurate conclusion is reached on Line 425 "…radiocarbon dates can be extreme minimum age constraints on deglaciation."

*AR:* **We can change the wording for more clarification here: "According to this interpretation, our radiocarbon ages from Unit 2 could reflect plant death anytime between moraine deposition and kettle basin stabilization."**

Line 366: Consider revising. "Ice cored moraines remained *as such…*" or "Moraines can remain ice cored for…"

*AR:* **Thanks, we will correct to 'Moraine can remain ice cored for…'**

**R3:**

Line 99/100: the 13,750-15,250 cal yr BP is from wood sample. This is an important point that may be more discussed later in the discussion section. Is the age calibrated against IntCal20 as your dataset? If not, it should be recalibrated and compared. This is also a general comment for all radiocarbon ages presented in the paper.

*AR:* **We will include this in the discussion around Line 365. The age is from wood within a marl layer that was deposited in a pond, so it is another basal radiocarbon age from a lake deposit and supports our conclusions. All radiocarbon ages in the text were recalibrated with IntCal20 and will be mentioned on Line 167.**

Line 204: Please justify here why you applied a MAM age (bleaching problem) and also which model was used  (MAM-3 or MAM-4 ?). What is the σb value used to calculate your MAM model? Also, even with a MAM age you may always over-estimate the depositional ages.

*AR:* **We will add a clause here that we and other studies in glaciofluvial environments use MAM's because of the increased potential for incomplete bleaching from subglacial or turbid water sediment transport that can sometimes shield sediment from complete bleaching. We will include more specifics with the MAM in the results section.**

Line 224 from 247: I recommend presenting the data by units but also by separating Kent moraine and Lake escarpment moraine sites. They are far away from each other… In my opinion, correlating data and units separated by more than 50 km is risky.

To avoid any confusions, you should write a paragraph for the Kent moraine sites and the different cores and then another one for the Lake escarpment site.

**Chronology section should be organized as in figure 5 :**

- Kent moraine with ages from VIN1 to VIN4, SONG1, 15ABB7

- Lake escarpment ages with 13DFK1 and 21LPB1

*AR:* **This comment is in alignment with comments by other R's and will be addressed with a new results section (see above).**

Line 265 : "*For 20VIN3, 20VIN4, and 21SONG1, the basal ages cluster around 14,700 cal yr BP*". This sentence should be placed at the end of the section after a detailed review of the ages.

How do you calculate the mean age of 14,700 yrs? Did you use an oxcal model to determine a pdf age?

*AR:* **The word 'cluster' was a non-technical term that describes the general agreement within uncertainty around 14,700 cal yr BP. We will replace this sentence with the details of each core in this new chronology section described in the general comments.**

Line 265 : SONG1-age is taken from Unit 3 whereas VIN-4 and VIN-3 ages are from unit 2. Why are you mixing ages from different units? Your ages are maybe "basal" but are in different stratigraphic units. Please justify.

**AR: We will include the Unit information for each radiocarbon sample in the new chronology section described in the general comments. The reason they are in different units is the availability of material for dating. We will clarify in the text that ages in Unit 2 are used as minimum-limits on deglaciation, and Unit 3 ages are used as minimum-limits on kettle basin formation.**

Line 266: "*The basal ages from the Lake Escarpment Moraine are 15,000-15,400 and 16,650-17,350 cal yr BP*." Again, I strongly recommend not mixing here the two sites. This sentence and all data from Lake escarpment should be placed together, in another paragraph.

*AR:* **See new strat/chronology section described in the general comments.**

Line 267 : "*The basal ages are not the oldest ages, however*". Please delete this sentence or rephrase it.

**AR: This will be rephrased in the discussion section.**

Line 270 : "Combined macrofossils …" : this is an important information that is not highlighted in the text and in figure 5. You should draw a different symbol for combined-fossils ages in figure 5, not only use stars. How many fossils are combined? Are they terrestrial or lacustrine?

*AR:* **In Unit 2, the sediment is very minerogenic, but millimeter- to sub-millimeter-sized macrofossils were present. Aiming for 2 mg of dry sample often meant 10+ pieces were combined. Please see response to R2 for terrestrial vs lacustrine samples. We will change the symbol for combined macrofossil vs full macrofossils in Figure 5 and the supplementary age-depth plots.**

Line 271 : "*In 20VIN3, the basal age is 14,350-15,150 cal yr BP, yet combined macrofossils higher in the core, at the Unit 2/3 boundary, produce an age of 15,350-15,650 cal yr BP.*" Again, these ages should go with the kent moraine.

Your basal age is from unit 2 and not from unit 3.

Again, you have a combined macrofossils sample, should be drawn with a different symbol.

*AR:* **See new results section described in the general comments and response above to which Unit our ages belong to.**

Line 280 : the radial plots placed in Appendix may be placed on figure 4 on D and the field photo may be placed on Appendix.

*AR:* **We will switch the figures.**

Line 283 : Same comment made on line 204

Line 301 to 305 : I don't understand how you came to that conclusion. Please rephrase this part.

*AR:* **Noted that R3 found this writing unclear, will revise for clarity.**

Line 306 to 314 : again this paragraph is hard to read. Maybe some rewording may be good there.

*AR:* **Noted that R3 found this writing unclear, will revise for clarity.**

Line 317 : "*The OSL ages support our estimated age of 25 – 20 ka for the Kent Moraine from prior literature and affirms our confidence in the age assignments using correlations of dated features elsewhere*". The sentence should be placed at the end of the paragraph.

*AR:* **This was also noted by R1, and we will move the sentence to the end.**

Line 317 : "our estimated ages" : Why our? Please replace by the.

*AR:* **We will replace with 'the'.**

Line 317 : Also cite references for the "prior literature"

**AR: We will cite Glover et al. (2011), Corbett et al. (2017), Stanford et al. (2020) and Balco et al. (2009; 2002).**

Line 321 : You should remember that they are MAM ages.

**AR: We state that these ages are from a minimum-age model and why we use the MAM in the methods in Line 204.**

Line 322 : "*The basal ages, taken at face value, indicate the deposition of the Kent Moraine occurred shortly before ~15 ka; this does not agree with our OSL age or the regional correlations*". Why? Please develop in the main text this conclusion. It is not a problem for me that lacustrine conditions occurred after the deposition of the sediments dated with OSL. Again, your OSL ages may overestimate the true age.

*AR:* **Section 5.2 will be restructured to explain these arguments better. Also see response to R2.**

Line 324 : "contradicts the 17 ka age …". Please cite the references here for this age. Based on which dating method? OSL, 14C or cosmogenic?

*AR:* **The 17 ka age was the oldest basal radiocarbon age from the Lake Escarpment moraine. This will be made more clear in the restructured 5.2 section.**

Line 338 : "We derived": why derive? Use another word.

*AR:* **We will change to "The age of…"**

Line 339 : "fish bone": again a missing information in figure 5 : Another symbol should be used for this sample!

*AR:* **We will define this sample as aquatic in Figure 5 and use a different symbol in the Supplementary Age-Depth plots.**

Line 334 : "*The macrofossil-rich rip-up clast in 20VIN1 holds evidence for two important interpretations: 1) the landscape was ice-free and at least sparsely vegetated as early as 19,350-19,600 cal yr BP (consistent with our OSL ages suggesting ice sheet retreat by 19.8 ± 2.6 – 20.6 ± 2.9 ka), and 2) the landscape stored this long-dead vegetation for thousands of years before it was redeposited.*" This sentence is not in a good position in the text. I recommend placing the sentence on line 338 after "*trustworthy age of 14,350-15,150 cal yr BP*"

*AR:* **We do not believe line 338 is a better position for this sentence. The paragraph surrounding line 338 is describing the ages we use in our analysis, and the sentence above is part of the analysis. We will restructure section 5.2 for clarity.**

Line 415 : "*The tundra zone is overlain by an interval with high spruce and pine pollen; this is the lowest unit found in the other five records (Miller, 1973; Calkin and McAndrews, 1980). This is likely reflecting the new forest biome associated with warmer temperatures*". Not well placed, I recommend moving it at line 411 after "*complicates their interpretation.*"

*AR:* **We agree this sentence best fits on line 411 and will move it there.**

Line 428 : In 10Be dating you have potentially inheritance problems that may over-estimate the ages of moraines. The age gap needs to be looked at more carefully and is under-discussed in your paper.

*AR:* **We do not think this discussion is within the scope of our paper.**

Line 444: "*The stratigraphically lowest radiocarbon ages from Unit 3 in the Lake Escarpment Moraine kettle basins, which are 15,000-15,400 and 13,600-14,000 cal yr BP, pre-date the ~13.1 ka re-advance suggested by Young et al. (2020)* ».

And if all your radiocarbon ages were all reworked or contained some reservoir effects?

*AR:* **With the new results section, we hope it will be more clear that these two ages are from terrestrial macrofossils within Unit 3, which has conformable radiocarbon ages (as discussed in above replies). As such, we do not think the macrofossils in these units are**

**reworked, nor could they be significantly affected by a hardwater effect. We hope the new age-depth models will help visual the sample placement.**

Line 444: For the age of Young et al. :   please remember on which kind of sample is based the age, piece of wood? You must discuss more here the data in my point of view.

*AR:* **See reply to general comment from R1.**

Line 444: Also, on line 99 an age of 13 750 – 15 250 yr BP is based on a piece of wood. How do you reconcile your data with these ages? On figure 2 this age is found really close to your site E , and looking your LPB1 section the ages look mostly in agreement, right? Your basal ages are close to those published ages. This may help…

*AR:* **See reply to previous comment from line 99.**

Line 460 : Again the 5 kyr offset could be due to some unbleached sediments, you can not totally delete this option.

*AR:* **The MAM has been found to successfully date glaciofluvial sediments with some portion of partial bleaching in other glacial settings in the northeast (Rittenour et al., 2015), and we believe this technique is working well in our study area. Our confidence is bolstered by the reworked macrofossils that date to 19 cal ka BP and the agreement in correlations to dated moraines in Ohio and eastern New York. We will include this wording in our discussion near Line 347.**

*RC3 Figures :*

Figure 3: Please indicate the core's names in the insets close to the colored dots. It is hard to follow the position of the cores and the descriptions in figure 5 when you are not familiar with the area.

*AR:* **We will add the core names next to the site names in the inset maps.**

Figure 5:

Please use different symbols according to the samples (terrestrial, lacustrine, combined macrofossils, fish bone…)

A chronostratigraphic model with spectra may be much better than just calibrated ages.

*AR:* **We will include different symbols for different samples in Figure 5. We will add age-depth plots in the supplement to include another way of viewing the radiocarbon ages.**

---

## Author Comment (AC2)

We appreciate the detailed and thorough reviews from Referee 1, 2 and 3. Between the three
reviews, there are common themes, which we group together and respond to at once when
appropriate. Below are responses to 1) their general comments, and 2) line-by-line comments.
We also include our response to the editor's comment on our initial revision.

The reviewer comments are unbolded and marked as Referee 1 (*R1*), Referee 2 (*R2*), and
Referee 3 (*R3*). The author's responses are marked as *AR* and **bolded**. This same document is
included in all three replies and responds to all referee and editorial comments. The line
numbers refer to the manuscript with tracked changes and are stated below the comment they
address.

*Presentation and discussion of the Allerød Re-advance Hypothesis*

*Referee 1 (R1):* 2) The "controversial" hypothesis should be presented in more detail in the
introduction or geological setting. Now it is only briefly mentioned in the introduction and again
in the discussion. It is very relevant to describe in detail how Young et al concluded that a
readvance took place at 13 ka.

*Author Response (AR):* **We will update the introduction to include the following text in**
**red:**

**"However, Young et al. (2020) recently interpreted new and existing radiocarbon ages**
**from western New York to support a significant re-advance of the LIS at ~13 ka that**
**overtopped the Lake Escarpment Moraine and nearly reached the Kent Moraine (Fig.**
**1). The evidence includes the re-interpretation of several unrelated sites throughout**
**western New York, but largely hinges on new trenched sections near the Kent Moraine**
**revealing logs in clayey diamicton, which Young et al. (2020) suggest requires glacial**
**overriding of a forest ~13.3 to ~13.0 ka. In contrast to Young et al.'s (2020) reconstruction,**
**most literature places the LIS margin north of Lake Ontario at this time (Dalton et al.,**
**2020; Muller and Calkin, 1993; Terasmae, 1980; and references therein), with the drainage**
**of Glacial Lake Iroquois occurring at ~13 ka (Fig. 1; Cronin et al., 2012; Lewis and**
**Anderson, 2019; Rayburn et al., 2005). To reconcile the disagreement in timing between**
**the hypothesized Allerød re-advance and existing chronologies, Young et al. (2020)**
**invoke a largely floating ice mass that left minimal traces of its existence in most areas. If**
**a re-advance of the scale hypothesized by Young et al. (2020) occurred (henceforth**
**referred to as the 'Allerød re-advance hypothesis'), we would need to revisit many**
**regional deglaciation chronologies.**

**Please see lines 56 - 67 with this updated text.**

*Presentation, organization, and discussion of the stratigraphy:*

*R2:* Section 4.1. It's easy to get lost in the stratigraphic descriptions for each core in Section 4.1.
It could be helpful for the reader to explicitly discuss each core based on the associated
moraine, so it becomes easier to follow when the stratigraphy of a Lake Escarpment moraine
core is being described versus a Kent moraine core.

*R3:* The *stratigraphy* and *sediment core chronology* sections need some deep reworking. Indeed, I found these sections poorly organized, making it difficult to read and understand the results. The different cores are mixed in the text and the different units are not well defined. A better presentation of the data is important, as the position of the samples within the units is subsequently used extensively to correlate ages. Suggestions for improvement are included in my comments line by line in the "specific comments".

**AR: These two comments suggest a reformatting of the results section to group the stratigraphy and radiocarbon results by sediment core and moraine. We will format the results sections to discuss each site in order by moraine, describing each sediment core individually. This individual description will include 1) descriptions of the core stratigraphy by Unit, 2) radiocarbon ages with information about the unit they come from and 3) the age-depth relationship.**

**Please see lines 248 - 307 for the updated text.**

*Presentation, organization, and discussion of the radiocarbon:*

*Associate editor:* Concerning the table 2 and reporting all calibrated intervals associated to the relative probability. I'll stick on the recommendations of the 14C community (e.g. Millard 2014). By reporting the interval that encompasses all calibrated intervals, you miss important information: the time periods that are unlikely and you no longer report a 95% confidence probability but something higher, between 0.95 and 1 since you include part of the remaining 5%. This is not mathematically correct. I would also draw your attention to the fact that the median makes no sense in the context of a multiinterval calibration. You may even end up with a median in a non-probable interval. You'll save space by eliminating this column. Furthermore, as the d13C are reported with uncertainty of 0.1‰, a single digit will sufficient to report this value. The spare room can be used to report 14C lab code. I understand your desire for an elegant table. My recommendation is to play with the thickness and color of the lines.

**AR: In Table 2 we will report the discrete solutions within the 95% confidence interval (as shown in your example table)  instead of the min and max of this range (as it is reported now). We will truncate the d13C to one decimal point.**

**Please see line 361 for the updated Table 2.**

*R1:* 1) The radiocarbon ages a given as ranges (min-max) throughout the text. Although this is the most correct way of reporting radiocarbon ages it makes the text less readable. I suggest that the min-max ranges are provided in the table and calibrated ages (in kiloyears) are used in the manuscript.

**AR: If the editor agrees, we would change the age presentation in the text and figures to median ages reported from Calib and uncertainties as the larger difference between the median and the maximum and minimum age, like this: "X.X ± X.X cal ka BP". Table 2 will still list ages in min-max form including discrete intervals as described above. Otherwise we will keep the age presentation in the text as the full 2-sigma range, as it is now, and refer to the table for more detailed information.**

**As advised from the associate editor, we report the 95% interval in the text as a range of ages in cal BP.**

*R1:* 3) It would be worthwhile to consider making age-depth models for the records where there are many radiocarbon ages. This would allow a better assessment of the potential outliers mentioned in the text and also plot the proxy data on an age scale.

*R3:* Radiocarbon ages should be placed in a chronological framework using chronostratigraphic models (in figure 5 or in the appendix) instead of just reported ages in a stratigraphic column as discussed in the main text. This should be easily done with Oxcal for example (using sequence or phase command…). Then we may have access to probability spectra, for individually calibrated and modelled ages. This would make it possible to better justify/approve the choices made by the authors to conclude that the ages are not those expected. This also allows them to better constrain the ages between the bottom and the top of the different cores. Maybe the radiocarbon ages will perhaps also be less rejected by the authors…

*AR:* **These two comments both suggest making age-depth models for the cores where we have sufficient radiocarbon constraints, so we will address their comments together. We agree that the age-depth plots provide a nice framework for discussing the radiocarbon results. We will add a supplemental file to our paper that contains age-depth plots so readers can visualize the sample distribution. We find the data shown as age-depth plots is most useful because we can symbolize the data by single terrestrial radiocarbon sample vs combined macrofossils with aquatic influence, etc., whereas an age-depth model created from rBacon or OxCal cannot. The 2σ age range is typically smaller than the symbol to show the sample age when viewing all the dates on a single plot, and the probability spectra for each age can be created with the raw data should a reader want to access this. Finally, our interpretation that Unit 2 records the collapse of ice-cored moraines and the creation of kettles in an unstable environment makes us hesitant to create an age-depth model through these sediments.**

**Please see the new Supplemental File for these age-depth plots. They are referenced in the text throughout the new results section (lines 248 - 307)**

*R2:* Section 4.2. It could be helpful to declare the stratigraphic unit in which each basal age is found since that is not consistent across all cores. As an example, starting on line 264 the basal ages for the Kent moraine are described, yet the 15,050-15,550 cal yr BP ages from 20VIN1 are in Unit 2 just above the boundary with Unit 1, whereas the 13,00-14,050 cal yr BP ages from 15ABB7 are found in Unit 3. Clarifying this for all radiocarbon data in Section 4.2 will strengthen your argument and allow the reader to immediately associate the ages with their stratigraphic unit, at the same time setting up the stratigraphic discordance with older ages higher up the core.

*AR:* **This will be clarified in the new structure of the results section.**

**Please see lines 248 - 307 for the updated text.**

*R2:* Section 5.2. The second paragraph of Section 5.2 discusses the basal radiocarbon ages in relation to the timing of LIS retreat from the Kent moraine. It would be helpful to clarify you are referring to the ages from 20VIN1 located in Unit 2 just above Unit 1 which is interpreted as the glacial till. The description of "shortly before ~15 ka" needs to be changed to "prior to ~15 ka" as these are minimum-limiting ages on deglaciation and the duration between deglacial onset, as indicated as the stratigraphic change from Unit 1 to Unit 2, and the radiocarbon ages cannot be determined from these data alone. As such, there is not necessarily a contradiction between the young basal radiocarbon ages and regional correlations with the Lake Escarpment moraine ~17 ka. These results simply state that abandonment of the Kent moraine happened prior to ~15 ka and must have occurred prior to deposition of the Lake Escarpment moraine ~17 ka. Additionally, this interpretation of the basal radiocarbon age lends support to the eventual age-lag conclusion due to persistent ice and permafrost within the moraine.

*AR:* **We agree that the nature of minimum-limiting radiocarbon ages means the ages are not contradictory to any of the correlations or the OSL ages. We will clarify wording according to this suggestion. A couple of line-by-line comments from R3 also suggest clarifying our discussion section that we address in more detail below.**

> **We removed this paragraph and replaced the information in lines 439 - 457. We find this new text a more straightforward way of discussing the minimum-limiting nature of each Unit. We also changed the wording in the abstract (Lines 22 - 29).**

*Methods section:*

*R2:* Section 3. There are no issues with the content of this section. However, many of the sentences start with the word "we" (e.g., "We collected…", "We determined…", "We returned…", etc.) often in sequential sentences.  I recognize that there is debate within the scientific community about the use of active or passive voice in writing, but for now the Methods section would read more fluidly if many of these sentences were changed to "Samples collected were analyzed for…" or "OSL analysis was conducted at…".

*AR:* **We appreciate this comment aimed at streamlining our writing. That said, this comment is a bit subjective (first vs. third person a matter of writers' preference) and would like this in active voice.**

*Line-by-line comments:*

**R1:**

Line 15: consider using another word than "provocative" - it gives the wrong impression.  Maybe use an alternative instead of provocative.

*AR:* **We agree to change the adverb provocative, perhaps to 'controversial', as described above.**

> **We chose to remove the adverb altogether (Line 15).**

Line 19: ..luminescence ages..."of what?

*AR:* **We will include that these ages are from topset beds in an ice-contact delta.**

**This information is included on Line 19.**

Line 29: see the first comment about "provocative"

*AR:* **See above reply.**

Line 50: More information is needed about how Young interpreted the radiocarbon ages (see
general comment)

*AR:* **See above reply.**

Line 69 (Fig. 1): Nice figure. Maybe add the existing chronological constraints mentioned in the
text.

*AR:* **We will add in data discussed in the text as points along the moraines.**

**Figure 1 has been updated with the dates discussed in the text (Line 80) and the
caption includes the citations (Lines 90-95).**

Line 173 (Fig 3): Nice figure but consider changing the yellow colour or increasing the line
thickness.

*AR:* **We will increase the line thickness.**

**Please see updated figure on Line 191 with increased line thickness.**

Line 214 (Fig. 4): Overall good figure but panel A could be improved. The dimensions seem off
and I wonder what the light brown colour below (a) represents.

*AR:* **R3 also has comments on how to improve this figure. We will sub-out the image in
Figure 4, check the alignments and dimensions, and provide in the figure that the light
brown is also bedrock.**

**Please see updated figure on Line 234 with fixed dimensions and bedrock labeled.**

Line 224: Ok descriptions, but it would be good to include the proxy data more in the description
of the 3 units. In particular, MS, CaCO3 and water content could be better incorporated in the
text.

*AR:* **We believe these data are best shown in the figures – we will refer to the graphs in
the beginning of the results sections to guide readers.**

**We refer to the graphs and broadly describe the downcore data on Lines 257 - 260.**

Line 249 (Fig. 5): Good summary figure with proxy data from the sediment cores. In 15ABB7
MS is 0 - is that a mistake? Also, some LOI and water content data are missing in 20VIN4.

*AR:* **Yes ABB7 has MS values of zero for the entire core. We did not measure LOI and
water content in 20VIN4 because it was a diamicton and difficult to sample, and we will
add that information into the figure.**

**Please see updated figure on Line 332.**

Line 317-321: Consider starting with summarizing new data before stating it supports the existing data.

*AR:* **This was also mentioned by R3 – we will move this sentence to the end of the paragraph.**

**We moved this sentence to the end of the paragraph (Lines 410 - 413).**

**R2:**

*Line specific comments:*

Line 36: Consider revising. "Well constrained ice sheet chronologies…constrain". Refine?

*AR:* **We would change the wording to: '…are necessary to *determine* the timing of …".**

**We changed this wording, see Line 43.**

Line 93: Consider revising. "…dated to…by radiocarbon dating" is redundant.

*AR:* **We would change the wording to: '…basin *around* 17 – 16 cal ka BP *based on* radiocarbon dating…'.**

**We changed this wording, see Line 108.**

Line 116: Define "significant" if you plan to describe the re-advance in this way.

*AR:* **Similar to R1 comments about 'provocative', we will revise.**

**We removed this adverb as well (Line 132).**

Line 154: Was sediment bulk density only measured on the Little Protection cores? If so, why not on the others?

*AR:* **The data are only from Little Protection because we investigated the Allerød re-advance in our two cores from the Lake Escarpment Moraine, and Dragonfly Kettle data creation took place before the Allerød re-advance hypothesis was published and we did not measure bulk density. We will include this information on Line 155.**

**Please see this new text in Lines 171 - 173.**

Line 283: Missing an end parenthesis after Olley et al. (1999).

*AR:* **Thanks! We will correct that.**

**Corrected in Line 370.**

Line 290: Consider using consistent terminology when discussing cores. Here you describe Vincent-1 when previously these cores were described as "20VIN1". You could also include the associated cores in parentheses after "Vincent-1" for clarity.

*AR:* **We will include the core name in parentheses after Vincent-1 for clarity. We will do the same if there is another occurrence of this.**

**We included the core name on Line 377.**

Line 300: Careful with the word "probably". This is an interpretation based on stratigraphy alone. A safer word to use here is "potentially".

*AR:* **We will change this to potentially.**

**We changed this wording on Line 388.**

Line 301: Reiterate that core 21LPB1 is associated with the Lake Escarpment moraine.

**AR: We will clarify which moraine the core is from within this section.**

**We include which moraine the cores are from in Line 387 and Line 389.**

Line 330: Clarify that "These samples…" refers to the macrofossils.

**AR: Thanks, we will clarify that.**

**We included this on Line 422.**

Line 336: Greater description is needed for how samples were assumed to be terrestrial origin. What visual cues were looked for to identify terrestrial vs. aquatic samples.

*AR:* **Identification was rare at the time of sampling, partly due to the small size of the macrofossils available to be collected. Dr. Ole Bennike identified some of the dated samples. We opted for measurements of $\delta^{13}C$ to provide a basis to infer terrestrial vs aquatic nature of samples dated. The samples that we identified as likely aquatic material had identifiable spores of aquatic material and the samples inferred to be terrestrial do not. So, we will change this sentence to be 'We move forward using samples assumed to be terrestrial from a lack of identifiable aquatic macrofossils and supported by $\delta^{13}C$ values'.**

**We include this new sentence in Lines 430-431.**

Line 348: Consider revising. The basal radiocarbon ages are trustworthy, but the up-core ages exhibit stratigraphic discordance and therefore do not reflect an accurate age of sediment deposition.

*AR:* **The Unit 2 ages are trustworthy as minimum-limiting constraints on moraine abandonment, but the evidence for slumps and rip-up clasts in Unit 2, plus the stratigraphic discordance in radiocarbon ages, are reasons to doubt the reliability of radiocarbon ages to reflect the age of the sediment they are within. We will include these reasons within this paragraph (Line 348) to clarify.**

**As described above, we have changed wording in the discussion section to better our interpretation of the ages in each Unit. Please see Lines 439 - 457.**

Line 364: The radiocarbon ages are from sediments stratigraphically above the glacial deposits therefore would not reflect moraine deposition. Do you mean "do not record deglacial onset"? A more accurate conclusion is reached on Line 425 "…radiocarbon dates can be extreme minimum age constraints on deglaciation."

*AR:* **We can change the wording for more clarification here: "According to this**
**interpretation, our radiocarbon ages from Unit 2 could reflect plant death anytime**
**between moraine deposition and kettle basin stabilization."**

**We have included this in Line 475.**

Line 366: Consider revising. "Ice cored moraines remained *as such…*" or "Moraines can remain
ice cored for…"

*AR:* **Thanks, we will correct to 'Moraine can remain ice cored for…'**

**Corrected on Line 479.**

**274 *R3:**

Line 99/100: the 13,750-15,250 cal yr BP is from wood sample. This is an important point that
may be more discussed later in the discussion section. Is the age calibrated against IntCal20 as
your dataset? If not, it should be recalibrated and compared. This is also a general comment for
all radiocarbon ages presented in the paper.

*AR:* **We will include this in the discussion around Line 365. The age is from wood within a**
**marl layer that was deposited in a pond, so it is another basal radiocarbon age from a**
**lake deposit and supports our conclusions. All radiocarbon ages in the text were**
**recalibrated with IntCal20 and will be mentioned on Line 167.**

**We have included that Nichols Brook acts as another example of delayed kettle**
**formation on Lines 475 - 477. We have stated that all ages in the text have been**
**recalibrated with IntCal20 (Line 186).**

Line 204: Please justify here why you applied a MAM age (bleaching problem) and also which
model was used  (MAM-3 or MAM-4 ?). What is the σb value used to calculate your MAM
model? Also, even with a MAM age you may always over-estimate the depositional ages.

*AR:* **We will add a clause here that we and other studies in glaciofluvial environments use**
**MAM's because of the increased potential for incomplete bleaching from subglacial or**
**turbid water sediment transport that can sometimes shield sediment from complete**
**bleaching. We will include more specifics with the MAM in the results section.**

**We have included why we use the MAM on Lines 223 - 225.**

Line 224 from 247: I recommend presenting the data by units but also by separating Kent
moraine and Lake escarpment moraine sites. They are far away from each other… In my
opinion, correlating data and units separated by more than 50 km is risky.

To avoid any confusions, you should write a paragraph for the Kent moraine sites and the
different cores and then another one for the Lake escarpment site.

# Chronology section should be organized as in figure 5 :

● Kent moraine with ages from VIN1 to VIN4, SONG1, 15ABB7

● Lake escarpment ages with 13DFK1 and 21LPB1

*AR:* **This comment is in alignment with comments by other R's and will be addressed**
**with a new results section (see above).**

**Please see new results section from Lines 248 - 307.**

Line 265 : "*For 20VIN3, 20VIN4, and 21SONG1, the basal ages cluster around 14,700 cal yr*
*BP".* This sentence should be placed at the end of the section after a detailed review of the
ages.

How do you calculate the mean age of 14,700 yrs? Did you use an oxcal model to determine a
pdf age?

*AR:* **The word 'cluster' was a non-technical term that describes the general agreement**
**within uncertainty around 14,700 cal yr BP. We will replace this sentence with the details**
**of each core in this new chronology section described in the general comments.**

**Each cores lowest age is now discussed individually in the new results section**
**(Lines 248 - 307).**

Line 265 : SONG1-age is taken from Unit 3 whereas VIN-4 and VIN-3 ages are from unit 2. Why
are you mixing ages from different units? Your ages are maybe "basal" but are in different
stratigraphic units. Please justify.

**AR: We will include the Unit information for each radiocarbon sample in the new**
**chronology section described in the general comments. The reason they are in different**
**units is the availability of material for dating. We will clarify in the text that ages in Unit 2**
**are used as minimum-limits on deglaciation, and Unit 3 ages are used as minimum-limits**
**on kettle basin formation.**

**The new results section outlines the Unit that each age is from (Lines 248 - 307)**
**and the next discussion section describes the interpretation of the radiocarbon**
**ages in each Unit (Lines 439 - 457).**

Line 266: "*The basal ages from the Lake Escarpment Moraine are 15,000-15,400 and*
*16,650-17,350 cal yr BP.*" Again, I strongly recommend not mixing here the two sites. This
sentence and all data from Lake escarpment should be placed together, in another paragraph.

*AR:* **See new strat/chronology section described in the general comments.**

**The sites are now described individually in the new results section (Lines 248 -**
**307)**

Line 267 : "*The basal ages are not the oldest ages, however".* Please delete this sentence or
rephrase it.

**AR: This will be rephrased in the discussion section.**

**This sentence was deleted when we rewrote the results section.**

Line 270 : "Combined macrofossils …" : this is an important information that is not highlighted in
the text and in figure 5. You should draw a different symbol for combined-fossils ages in figure 5,
not only use stars. How many fossils are combined? Are they terrestrial or lacustrine?

*AR:* **In Unit 2, the sediment is very minerogenic, but millimeter- to sub-millimeter-sized**
**macrofossils were present. Aiming for 2 mg of dry sample often meant 10+ pieces were**
**combined. Please see response to R2 for terrestrial vs lacustrine samples. We will**
**change the symbol for combined macrofossil vs full macrofossils in Figure 5 and the**
**supplementary age-depth plots.**

**Please see figures 5a, 5b, and all the supplementary files for updated symbols**
**showing the difference between radiocarbon sample types and lines 177-179 and**
**430-431 for macrofossil identification text.**

Line 271 : "*In 20VIN3, the basal age is 14,350-15,150 cal yr BP, yet combined macrofossils*
*higher in the core, at the Unit 2/3 boundary, produce an age of 15,350-15,650 cal yr BP.*" Again,
these ages should go with the kent moraine.

Your basal age is from unit 2 and not from unit 3.

Again, you have a combined macrofossils sample, should be drawn with a different symbol.

*AR:* **See new results section described in the general comments and response above to**
**which Unit our ages belong to.**

Line 280 : the radial plots placed in Appendix may be placed on figure 4 on D and the field
photo may be placed on Appendix.

*AR:* **We will switch the figures.**

**Panel (d) has been switched to show the radial plots (Line 234).**

Line 283 : Same comment made on line 204

Line 301 to 305 : I don't understand how you came to that conclusion. Please rephrase this part.

*AR:* **Noted that R3 found this writing unclear, will revise for clarity.**

**We chose to remove the specific details in Lines 390-393 and describe the**
**interpretation more broadly like we did for 20VIN4 (Line 387-388).**

Line 306 to 314 : again this paragraph is hard to read. Maybe some rewording may be good
there.

*AR:* **Noted that R3 found this writing unclear, will revise for clarity.**

**We streamlined this paragraph to include the most important points and use more**
**clear language. Instead of 'productive lake and landscape' we changed the**
**wording to 'more vegetation growing in the lake and landscape'. We removed the**
**sentence about minerogenic sediment layers because it was redundant as we**
**subsequently discuss the rip-up clasts later in the paragraph. (Lines 394 - 403).**

Line 317 : "*The OSL ages support our estimated age of 25 – 20 ka for the Kent Moraine from*
*prior literature and affirms our confidence in the age assignments using correlations of dated*
*features elsewhere*". The sentence should be placed at the end of the paragraph.

*AR:* **This was also noted by R1, and we will move the sentence to the end.**

**This was moved to the end of the sentence (Lines 410 - 413).**

Line 317 : "our estimated ages" : Why our? Please replace by the.

*AR:* **We will replace with 'the'.**

**Replaced with 'the', line 410.**

Line 317 : Also cite references for the "prior literature"

**AR: We will cite Glover et al. (2011), Corbett et al. (2017), Stanford et al. (2020) and Balco**
**et al. (2009; 2002).**

**These citations are included (Line 412-413).**

Line 321 : You should remember that they are MAM ages.

**AR: We state that these ages are from a minimum-age model and why we use the MAM in**
**the methods in Line 204.**

**Our MAM explanation is now placed in Lines 223 - 225.**

Line 322 : "*The basal ages, taken at face value, indicate the deposition of the Kent Moraine*
*occurred shortly before ~15 ka; this does not agree with our OSL age or the regional*
*correlations*". Why? Please develop in the main text this conclusion. It is not a problem for me
that lacustrine conditions occurred after the deposition of the sediments dated with OSL. Again,
your OSL ages may overestimate the true age.

*AR:* **Section 5.2 will be restructured to explain these arguments better. Also see response**
**to R2.**

**Restructure is found in lines 439 - 457.**

Line 324 : "contradicts the 17 ka age …". Please cite the references here for this age. Based on
which dating method? OSL, 14C or cosmogenic?

*AR:* **The 17 ka age was the oldest basal radiocarbon age from the Lake Escarpment**
**moraine. This will be made more clear in the restructured 5.2 section.**

**This was deleted in the restructuring.**

Line 338 : "We derived": why derive? Use another word.

*AR:* **We will change to "The age of…"**

**This was deleted in the restructuring.**

Line 339 : "fish bone": again a missing information in figure 5 : Another symbol should be used
for this sample!

*AR:* **We will define this sample as aquatic in Figure 5 and use a different symbol in the**
**Supplementary Age-Depth plots.**

**See Figure 5 on Line 334 and Supplementary Figure 5.**

Line 334 : "*The macrofossil-rich rip-up clast in 20VIN1 holds evidence for two important*
*interpretations: 1) the landscape was ice-free and at least sparsely vegetated as early as*
*19,350-19,600 cal yr BP (consistent with our OSL ages suggesting ice sheet retreat by 19.8 ±*
*2.6 – 20.6 ± 2.9 ka), and 2) the landscape stored this long-dead vegetation for thousands of*
*years before it was redeposited.*" This sentence is not in a good position in the text. I
recommend placing the sentence on line 338 after "*trustworthy age of 14,350-15,150 cal yr BP*"

*AR:* **We do not believe line 338 is a better position for this sentence. The paragraph**
**surrounding line 338 is describing the ages we use in our analysis, and the sentence**
**above is part of the analysis. We will restructure section 5.2 for clarity.**

**We believe this sentence is in a better place now that this paragraph is describing**
**how we interpret Unit 2 (Lines 439-449).**

Line 415 : "*The tundra zone is overlain by an interval with high spruce and pine pollen; this is*
*the lowest unit found in the other five records (Miller, 1973; Calkin and McAndrews, 1980). This*
*is likely reflecting the new forest biome associated with warmer temperatures*". Not well placed, I
recommend moving it at line 411 after "*complicates their interpretation.*"

*AR:* **We agree this sentence best fits on line 411 and will move it there.**

**This sentence is now in Lines 524 - 527.**

Line 428 : In 10Be dating you have potentially inheritance problems that may over-estimate the
ages of moraines. The age gap needs to be looked at more carefully and is under-discussed in
your paper.

*AR:* **We do not think this discussion is within the scope of our paper.**

Line 444: "*The stratigraphically lowest radiocarbon ages from Unit 3 in the Lake Escarpment*
*Moraine kettle basins, which are 15,000-15,400 and 13,600-14,000 cal yr BP, pre-date the*
*~13.1 ka re-advance suggested by Young et al. (2020)* »*.

And if all your radiocarbon ages were all reworked or contained some reservoir effects?

*AR:* **With the new results section, we hope it will be more clear that these two ages are**
**from terrestrial macrofossils within Unit 3, which has conformable radiocarbon ages (as**
**discussed in above replies). As such, we do not think the macrofossils in these units are**
**reworked, nor could they be significantly affected by a hardwater effect. We hope the new**
**age-depth models will help visual the sample placement.**

Line 444: For the age of Young et al. :   please remember on which kind of sample is based the
age, piece of wood? You must discuss more here the data in my point of view.

*AR:* **See reply to general comment from R1.**

**We include more information on the Young data in the introduction (Lines 56 - 65)**
**and on lines 566 - 568 we reference that their data is primarily based on logs**
**within clayey diamicton.**

Line 444: Also, on line 99 an age of 13 750 – 15 250 yr BP is based on a piece of wood. How do
you reconcile your data with these ages? On figure 2 this age is found really close to your site E

, and looking your LPB1 section the ages look mostly in agreement, right? Your basal ages are
close to those published ages. This may help…

*AR:* **See reply to previous comment from line 99.**

Line 460 : Again the 5 kyr offset could be due to some unbleached sediments, you can not
totally delete this option.

*AR:* **The MAM has been found to successfully date glaciofluvial sediments with some**
**portion of partial bleaching in other glacial settings in the northeast (Rittenour et al.,**
**2015), and we believe this technique is working well in our study area. Our confidence is**
**bolstered by the reworked macrofossils that date to 19 cal ka BP and the agreement in**
**correlations to dated moraines in Ohio and eastern New York. We will include this**
**wording in our discussion near Line 347.**

**We mention that the 19,350 - 19,600 cal BP age lends confidence to our MAM on**
**Line 448.**

*RC3 Figures :*

Figure 3: Please indicate the core's names in the insets close to the colored dots. It is hard to
follow the position of the cores and the descriptions in figure 5 when you are not familiar with the
area.

*AR:* **We will add the core names next to the site names in the inset maps.**

**See Figure 3 on Line 191.**

Figure 5:

Please use different symbols according to the samples (terrestrial, lacustrine, combined
macrofossils, fish bone…)

A chronostratigraphic model with spectra may be much better than just calibrated ages.

*AR:* **We will include different symbols for different samples in Figure 5. We will add**
**age-depth plots in the supplement to include another way of viewing the radiocarbon**
**ages.**

**We have displayed the radiocarbon samples based on single vs combined,**
**terrestrial vs aquatic in Figure 5 on Line 335 and also in the Supplementary.**

---

## Author Comment (AC4)

[Figure]

Figure S1. 20VIN1 age-depth values plotted by depth and symbolized by terrestrial vs aquatic nature and single vs combined nature. Depth begins at the top of the sediment core and for this coring location that is the surface of the bog.

9

[Figure]

Figure S2. 20VIN3 age-depth values plotted by depth and symbolized by terrestrial vs aquatic nature and single vs combined nature. Depth begins at the top of the sediment core and for this coring location that is not the surface of the bog.

[Figure]

Figure S3. 20VIN4 age-depth values plotted by depth and symbolized by terrestrial vs aquatic nature and single vs combined nature. Depth begins at the top of the sediment core and for this coring location that is not the surface of the bog.

[Figure]

Figure S4. 15ABB7 age-depth values plotted by depth and symbolized by terrestrial vs aquatic nature and single vs combined nature.  Depth begins at the top of the sediment core and for this coring location that is not the surface of the bog.

[Figure]

Figure S5. 21LPB1 age-depth values plotted by depth and symbolized by terrestrial vs aquatic nature and single vs combined nature. The timing of the Allerød re-advance hypothesis is shown by the gray line. Depth begins at the top of the sediment core and for this coring location that is the surface of the bog.

[Figure]

Figure S6. 13DFK1 age-depth values plotted by depth and symbolized by terrestrial vs aquatic nature and single vs combined nature. The timing of the Allerød re-advance hypothesis is shown by the gray line. Depth begins at the top of the sediment core and for this coring location that is the surface of the bog.

Table S1. Dose Rate Information

| USU num. | Lat/Long | In-situ $H_2O$ (%) | $D_R$ Subsample[1] | K (%)[2] | Rb (ppm)[2] | Th (ppm)[2] | U (ppm)[2] | Cosmic (Gy/kyr) |
|---|---|---|---|---|---|---|---|---|
| USU-3622 | 42.11394/ -78.94899 | 7.5 | F: 70% | 1.64±0.04 | 77.6±3.1 | 7.8±0.7 | 2.6±0.2 | 0.18±0.02 |
| | | | M: 20% | 1.12±0.03 | 58.7±2.3 | 8.6±0.8 | 2.2±0.2 | |
| | | | C: 10% | 1.29±0.03 | 76.1±3.0 | 11.1±1.0 | 2.1±0.1 | |
| USU-3623 | 42.11394/ -78.94899 | 20.0 | F: 85% | 1.52±0.04 | 74.4±3.0 | 8.3±0.7 | 2.0±0.1 | 0.18±0.02 |
| | | | M: 15% | 1.35±0.03 | 72.7±2.9 | 8.4±0.8 | 2.4±0.2 | |

[1] Dose rate ($D_R$) subsamples based on grain size: fine-F (<1.7 mm), medium-M (1.7-16 mm), coarse-C (>16 mm), and weighted proportions (%) of subsamples used with chemistry in gamma dose rate calculation. Beta dose rate uses chemistry from fine fraction (<1.7 mm) only.

[2] Radioelemental concentrations determined using ICP-MS and ICP-AES techniques; dose rate is derived from concentrations by conversion factors from Guérin et al. (2011).

44

45

[Figure]

46

47 Figure S7. Example luminescence signal decay (left) and dose-response curves from 5 aliquots from each of the
48 luminescence samples.